# A junctional PACSIN2/EHD4/MICAL-L1 complex coordinates VE-cadherin trafficking for endothelial migration and angiogenesis

Tsveta S. Malinova[1,5], Ana Angulo-Urarte [1,5], Julian Nüchel [2], Marina Tauber[2], Miesje M. van der Stoel [1], Vera Janssen [1], Annett de Haan[1], Anouk G. Groenen[1], Merel Tebbens [1], Mariona Graupera [3,4], Markus Plomann [2] & Stephan Huveneers [1✉]

Angiogenic sprouting relies on collective migration and coordinated rearrangements of endothelial leader and follower cells. VE-cadherin-based adherens junctions have emerged as key cell-cell contacts that transmit forces between cells and trigger signals during collective cell migration in angiogenesis. However, the underlying molecular mechanisms that govern these processes and their functional importance for vascular development still remain unknown. We previously showed that the F-BAR protein PACSIN2 is recruited to tensile asymmetric adherens junctions between leader and follower cells. Here we report that PACSIN2 mediates the formation of endothelial sprouts during angiogenesis by coordinating collective migration. We show that PACSIN2 recruits the trafficking regulators EHD4 and MICAL-L1 to the rear end of asymmetric adherens junctions to form a recycling endosome-like tubular structure. The junctional PACSIN2/EHD4/MICAL-L1 complex controls local VE-cadherin trafficking and thereby coordinates polarized endothelial migration and angiogenesis. Our findings reveal a molecular event at force-dependent asymmetric adherens junctions that occurs during the tug-of-war between endothelial leader and follower cells, and allows for junction-based guidance during collective migration in angiogenesis.

---

[1] Department of Medical Biochemistry, Amsterdam Cardiovascular Sciences, Amsterdam UMC, University of Amsterdam, location AMC, Amsterdam, The Netherlands. [2] Center for Biochemistry, Faculty of Medicine, University of Cologne, Cologne, Germany. [3] Vascular Biology and Signaling Group, ProCURE, Oncobell Program, Institut d'Investigació Biomèdica de Bellvitge (IDIBELL), Gran Via de l'Hospitalet 199, 08908 L'Hospitalet de Llobregat, Barcelona, Spain. [4] CIBERONC, Instituto de Salud Carlos III, Madrid, Spain. [5] These authors contributed equally: Tsveta S. Malinova, Ana Angulo-Urarte. ✉email: s.huveneers@amsterdamumc.nl

Sprouting angiogenesis, the formation of new blood vessels that originate from pre-existing vasculature, is an essential process for development, wound healing, and tumorigenesis. Coordinated endothelial migration and rearrangements drive angiogenesis, and are important for related vascular developmental processes such as anastomosis, lumen formation, and valve morphogenesis[1–5]. Angiogenic sprouts are formed by migrating endothelial leader cells (tip cells), which are directly connected to endothelial follower cells (stalk cells), which mediate sprout elongation and vessel branching[6–8]. Endothelial cell–cell contacts are formed through VE-cadherin-based adherens junctions (AJs). AJ remodeling is needed to allow the formation of angiogenic sprouts from the vascular endothelium, whereas stabilization of AJs supports the integrity of newly formed vessels. Hence, a tight regulatory system that is responsible for the making and breaking of endothelial cell–cell contacts is needed for proper sprouting angiogenesis and vascular barrier function[9–11]. Although the importance of VE-cadherin-based AJs in angiogenesis has already been well recognized, the molecular events that explain the spatiotemporal turnover of AJs between leader and follower cells are still poorly understood.

VE-cadherin is a transmembrane receptor that connects neighboring cells by forming extracellular homotypic adhesions[11–14]. Intracellularly, VE-cadherin binds to the actin cytoskeleton via β-catenin and α-catenin proteins[11]. Besides its function as a structural anchor between endothelial cells and their actomyosin cytoskeleton, VE-cadherin possesses important mechanotransduction properties, which coordinate endothelial collective behavior[15–18]. Cytoskeletal-derived forces remodel the organization, protein composition, and function of AJs in cultured endothelial cells[15,16,19–21] and in the endothelium of developing blood vessels[4,7,22]. In turn, cytoplasmic molecular adaptations within the AJ complex are propagated over multicellular distances via the extracellular VE-cadherin connections[23,24]. Such mechanically coupled AJ signaling ensures the finely tuned collective behavior of the endothelium in response to pushing and pulling forces from neighboring cells[24–28].

During collective cell migration, actomyosin contractions occur at the rear of the leader cells, which pulls along the front of the follower cells. These pulling forces at the interface between leader and follower cells produce tension on the cell–cell contacts, leading to the formation of asymmetric AJs[14,29–31]. Tensile AJs are essential for collective dynamics in cell monolayers[26,27]. The polarized trafficking of cadherin-based adhesions is thought to drive this phenomenon[32]. The formation of asymmetric tensile AJs is accompanied by local curvature of the junctional plasma membrane[19,33]. We previously showed that recruitment of the curvature-sensing F-BAR protein PACSIN2 (protein kinase C and casein kinase 2 substrate in neurons 2) occurs specifically at the rear of asymmetric AJs in the follower cells[19]. Intriguingly, the formation of asymmetric AJs precedes the change in the direction of collectively migrating cells, suggesting that BAR proteins provide follower cells with guidance signals in vitro[14,33]. The identification of BAR proteins as molecular players at AJs added curvature sensing of the plasma membrane as contributor to mechanotransduction and junctional remodeling. The previous studies were performed with endothelial cell cultures, and although PACSIN2-positive asymmetric junctions have been observed in human blood vessels as well, to this date, the functional importance of junction-based signaling through BAR proteins in endothelial collective behavior and vascular development remains unknown. In addition, the molecular systems responsible for PACSIN2-driven remodeling of VE-cadherin at asymmetric junctions have not yet been identified.

Here, we reveal that junctional PACSIN2 signaling guides endothelial cells during angiogenesis. Our results show that PACSIN2 mediates endothelial front–rear polarity during collective migration by recruiting the trafficking regulators EHD4 and MICAL-L1 specifically to asymmetric AJs formed between endothelial leader and follower cells. The junctional PACSIN2/EHD4/MICAL-L1 complex controls VE-cadherin trafficking and coordinates endothelial migration and angiogenesis. Together, these data show that tensile asymmetric AJs harbor a plasma membrane-bound tubular recycling compartment with marked importance in AJ remodeling and control over endothelial collective migration in the developing vasculature. These findings provide a mechanistic explanation for the established hypothesis of long-range guidance communication between endothelial cells during angiogenesis.

## Results

**PACSIN2 controls sprouting angiogenesis in vivo.** The F-BAR protein PACSIN2 is recruited to the rear of AJs between endothelial leader–follower cells in vitro to locally protect junctional integrity[19]. In vitro cultured endothelial cells express the PACSIN2 and PACSIN3 isoforms, but no PACSIN1[19]. We first immuno-stained postnatal day 6 (P6) wild-type retinas and confirmed that PACSIN2, but not the related PACSIN3 isoform, is expressed in the endothelium of the developing vasculature of the retina in vivo (Supplementary Fig. 1a). Next, we generated homozygous $Pacsin2^{-/-}$ knockout mice through homologous recombination and ubiquitous Cre-loxP recombination (Supplementary Fig. 1b–d, see "Methods"). Lack of PACSIN2 protein expression was confirmed in the retina and lung tissue from the $Pacsin2^{-/-}$ mice. $Pacsin2$ gene deletion resulted in a slight increase in PACSIN1 and PACSIN3 protein levels in the overall $Pacsin2^{-/-}$ retinal tissue (Supplementary Fig. 1e). $Pacsin2^{-/-}$ mice are viable, fertile and adult mice appear healthy without major defects or bleedings, which indicates that there is no prominent vascular barrier defect. Some mild delays in cardiomyocyte development were previously reported in $Pacsin2^{-/-}$ mice[34]. To investigate the role of PACSIN2 during angiogenesis, we compared the developing vasculature in control ($Pacsin2^{+/+}$) and $Pacsin2^{-/-}$ mice retinas at P6. $Pacsin2^{-/-}$ P6 retinas did not show explicit differences in the number of developing vascular branches and sprouts at the vascular front (Fig. 1a, c). However, we noticed that the vascular sprouts were shorter, albeit composed by a higher number of endothelial cells (Fig. 1b–d). This coincided with the formation of abnormal angiogenic sprouts, constituted by endothelial clusters of three or more cells, instead of the clear leader and follower cells that generate sprouts in control vessels (Fig. 1b, d). $Pacsin2^{-/-}$ retinas showed no differences in the number of endothelial dividing cells (Fig. 1d). This suggests that the increase in endothelial cell numbers at the vascular sprouts is not due to proliferative changes, but might relate to differences in endothelial cell organization. Furthermore, we observed no changes in the number of endothelial tip cells at the sprouting front in $Pacsin2^{-/-}$ retinas, as defined by endothelial cell-specific molecule 1 (ESM1) staining (Fig. 1d and Supplementary Fig. 2a).

Endothelial cell migration and rearrangements in angiogenic sprouts requires the remodeling of the VE-cadherin-based AJs[4,7,35]. High-magnification imaging of VE-cadherin stainings revealed that the AJs in angiogenic vessels in the $Pacsin2^{-/-}$ retinas are differently organized (Fig. 1e). Notably, we observed an increase in cytoplasmic VE-cadherin intensity at the sprouting front in the $Pacsin2^{-/-}$ retinas (Fig. 1e, f), a feature which has been shown to associate with the remodeling of AJs between endothelial leader and follower cell during angiogenesis[7]. Moreover, the VE-cadherin junctions were completely disorganized in the large endothelial clusters at the sprouting front in the $Pacsin2^{-/-}$ retinas (Supplementary Fig. 2b). Taken together,

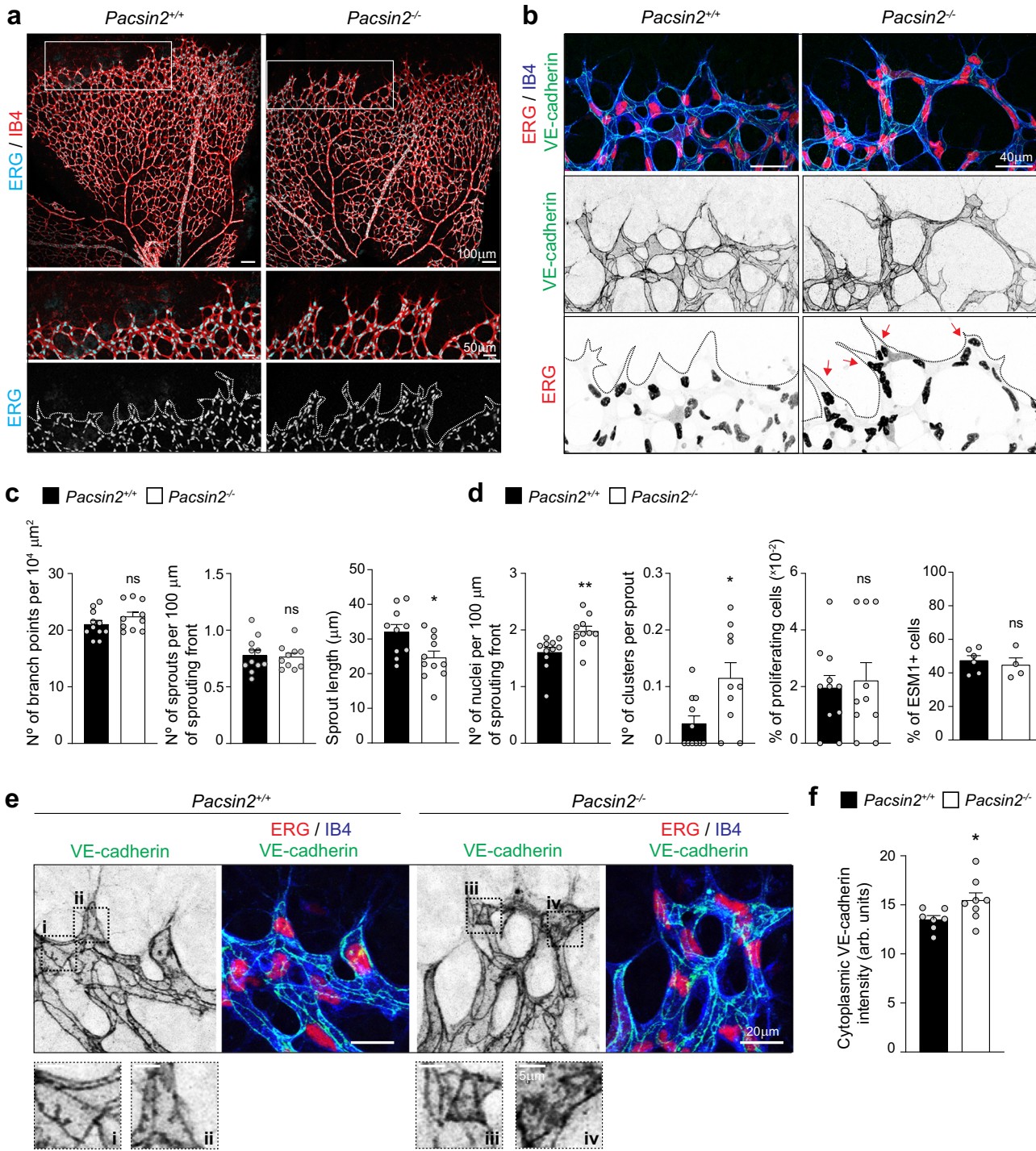

these findings show that PACSIN2 is needed for endothelial organization during sprouting angiogenesis and that PACSIN2 modulates VE-cadherin-based endothelial junctions in vivo.

**Endothelial PACSIN2 coordinates collective migration and angiogenic sprouting.** To understand how PACSIN2 controls endothelial cell movement, we examined in vitro cultured primary human umbilical vein endothelial cells (HUVECs) in which *PACSIN2* expression was depleted by specific short hairpin RNAs (shRNAs, previously validated in ref. [19]) (Fig. 2a). Multicellular spheroids composed of shControl or shPACSIN2 HUVECs were placed in 3D collagen matrices[36] to assess VEGF-induced sprouting capacity. Sprout formation and elongation from

spheroids was strongly decreased upon depletion of PACSIN2 (Fig. 2b, c), confirming that PACSIN2 is needed for endothelial-driven angiogenic sprouting. We next performed scratch wound assays of monolayers formed by shControl or shPACSIN2 HUVECs. Depletion of PACSIN2 inhibited cell migration toward the scratch resulting in delayed wound closure (Fig. 2d, e and Supplementary Movie 1). Moreover, applying particle image velocity (PIV) analysis[27,37,38] to the time-lapse recordings revealed that there is a strong decrease in the correlated migration of a given cell and its neighbors upon depletion of PACSIN2 (Fig. 2f). To decipher whether the depletion of PACSIN2 results in a cell-intrinsic or a collective migration defect, we performed competition scratch assays. Mosaic endothelial monolayers were

**Fig. 1 Abnormal sprouting during retinal angiogenesis in *Pacsin2*−/− mice. a** Immunofluorescent images of whole-mount retinas stained for Isolectin B4 (IB4, red, as marker of blood vessels) and the endothelial transcription factor ERG (cyan, marker of endothelial nuclei) from control (*Pacsin2*+/+) and *Pacsin2*−/− mouse littermates at P6. Representative images of at least ten retinas per genotype from three independent littermates. White rectangles indicate regions of interest (ROI) of which magnifications are shown below. **b** Representative images of the sprouting front of retinas from control and *Pacsin2*−/− mouse littermates at P6 stained for ERG (red), IB4 (blue), and VE-cadherin (green). Punctuated lines indicate the sprouting front boundary. The red arrows indicate nuclear clusters in abnormal sprouts. **c** Quantification of the number of branch points per unit of area, the number of sprouts per 100 μm of sprouting front border, and the average length of sprouts (at least $n = 10$ retinas per genotype, from three independent littermates). $P = 0.0295$ when comparing the sprout length in *Pacsin2*+/+ to *Pacsin2*−/− retinas. **d** Quantification of the number of nuclei per 100 μm of sprouting front border, the number of endothelial cell clusters per sprout, and the percentage of proliferating cells based on cytoplasmic (dividing cells) versus nuclear ERG (non-dividing cells) stainings per unit area in the sprouting front of control and *Pacsin2*−/− P6 retinas (at least ten retinas per genotype, from at least three independent littermates). $P = 0.0015$ when comparing the number of nuclei, and $P = 0.0242$ when comparing the number of endothelial clusters in *Pacsin2*+/+ to *Pacsin2*−/− retinas. Quantification of the number of tip cells (ESM1+) per total number of ECs at the sprouting front (at least four retinas per genotype, from two independent littermates). **e** Representative high-resolution images of the sprouting front from control and *Pacsin2*−/− mice stained for VE-cadherin (green), ERG (red), and IB4 (blue). The black dotted squares indicate the ROIs that are magnified in the panels below. **f** Quantification of the intensity of cytoplasmic VE-cadherin at the vascular sprouts of control and *Pacsin2*−/− retinas (at least seven retinas per genotype, from two independent littermates). $P = 0.0401$ when comparing *Pacsin2*+/+ to *Pacsin2*−/− retinas. All quantifications represent mean ± SEM (error bars) and the statistics were performed by the two-sided Mann–Whitney test. ns non-significant; *$P < 0.05$; **$P < 0.01$. Scale bars, 100 and 50 μm (**a**), 40 μm (**b**), 20 and 5 μm (**e**). Arb. units arbitrary units. Source data are provided as a Source Data file.

generated in which half of the HUVECs expressed shControl with an RFP tag and the other half of the HUVECs expressed shPACSIN2 and GFP. Next, scratch assays were performed and the identity of the front row cells during collective migration was determined. The experiments demonstrated that at 12 h after scratch the migrating front is dominated by cells that express PACSIN2, whereas PACSIN2-depleted cells fail to follow during the collective cell migration process (Fig. 2g–i). These findings indicate that PACSIN2 is needed for coordinated collective cell migration.

Next, we studied whether PACSIN2 might control the front–rear polarization of migrating ECs. Efficient collective migration is tightly dependent on the ability of the cells to polarize toward a directional cue[39,40], and in polarized migrating endothelial cells, the Golgi is located in front of the nucleus[41,42]. We assessed Golgi localization in the first two rows of scratch wound migrating shControl and shPACSIN2 HUVECs (Fig. 2j). The Golgi was considered oriented if localized within a 120° margin relative to the nucleus and the leading edge of the cells (Fig. 2k). These experiments demonstrate a strong reduction in cell polarization upon PACSIN2 depletion (Fig. 2l). To determine the importance of PACSIN2 for endothelial polarization during angiogenesis in vivo, we investigated Golgi orientation in ECs at the angiogenic front in control and *Pacsin2*−/− P6 retinas. *Pacsin2*−/− ECs were strongly impaired in their capacity to polarize toward the sprouting front (Fig. 2m–o). Taken together, these results indicate that PACSIN2 is needed for endothelial migration by supporting cell collectivity and polarity during angiogenesis.

**PACSIN2 recruitment occurs in parallel with the dissociation of p120-catenin from VE-cadherin.** Our previous study has shown that asymmetric AJs are formed between the leader and the follower cells during directed migration, and that PACSIN2 is recruited to the rear side of these contacts in the follower cells (Fig. 3a and ref. [19]). Since AJ turnover is essential for efficient collective migration[2,32,43–45], and PACSIN2 locally controls VE-cadherin dynamics at asymmetric AJs[19], we next investigated how PACSIN2 regulates VE-cadherin turnover. P120-catenin binds to the juxtamembrane domain of the VE-cadherin cytoplasmic tail where it masks an endocytic motif[46,47]. P120-catenin dissociation exposes this motif and induces clathrin-dependent endocytosis of the VE-cadherin receptor[48,49]. To establish whether PACSIN2 and p120-catenin are functionally associated, we performed immunofluorescence (IF) imaging of endogenous p120-catenin, PACSIN2, and VE-cadherin in HUVECs (Fig. 3b). Linescan

analysis across individual asymmetric AJs revealed that p120-catenin is released from the trailing ends of PACSIN2-positive, but not PACSIN2-negative, asymmetric AJs (Fig. 3c, d). In addition, live imaging of HUVECs expressing PACSIN2-GFP, p120-catenin-mCherry, and live-labeled VE-cadherin showed that PACSIN2 is recruited to the asymmetric AJs after p120-catenin dissociates from VE-cadherin (Supplementary Movie 2 and Supplementary Fig. 3a). Together these data demonstrate that PACSIN2 is recruited to the asymmetric AJs that are locally primed for internalization due to the release of p120-catenin from the VE-cadherin complex.

We next investigated if PACSIN2 is a member of the constitutive endocytic machinery recruited to VE-cadherin after p120-catenin dissociation[48,50]. First, endogenous VE-cadherin was depleted from HUVECs by lentiviral shRNAs targeting the 3′-untranslated region (3′-UTR) of the *CDH5* messenger RNA (Supplementary Fig. 3b). These VE-cadherin knockdown cells were rescued by ectopic expression of wild-type VE-cadherin-GFP, VE-cadherin[DEE]-GFP, or VE-cadherin[GGG]-GFP; two previously described VE-cadherin variants that contain mutations at the core p120-catenin binding site (Fig. 3e, f). Both VE-cadherin mutants do not bind to p120-catenin. However, the DEE mutation prevents endocytosis of VE-cadherin and affects endothelial polarization and retinal angiogenesis, while the GGG mutation still allows for VE-cadherin internalization[48,51]. IF imaging of the generated endothelial cell lines shows that recruitment of endogenous PACSIN2 to VE-cadherin-, VE-cadherin[DEE]-, and VE-cadherin[GGG]-based asymmetric AJs occurs at comparable levels (Fig. 3e, g). This indicates that junctional recruitment of PACSIN2 occurs in parallel with the mechanism driving p120-catenin controlled VE-cadherin endocytosis.

**PACSIN2, EHD4, and MICAL-L1 bind to each other at the trailing end of asymmetric AJs.** To identify the mechanism through which PACSIN2 controls VE-cadherin turnover, we next looked at potential PACSIN2 interactors involved in protein trafficking. PACSIN2 contains Asn-Pro-Phe (NPF) motifs that bind to the EH domain of the dynamin-like C-terminal Eps15 homology domain (EHD) proteins[52] (Fig. 4a). There are four known mammalian EHD proteins, which are linked to different branches of endocytic pathways[53–55]. IF imaging in HUVECs revealed that PACSIN2 colocalizes clearly with EHD4, and to a certain extent with EHD1, in a tubular fashion at the asymmetric AJs (Fig. 4b). No junctional colocalization between PACSIN2 and EHD2 or EHD3 was observed. Quantified Pearson's

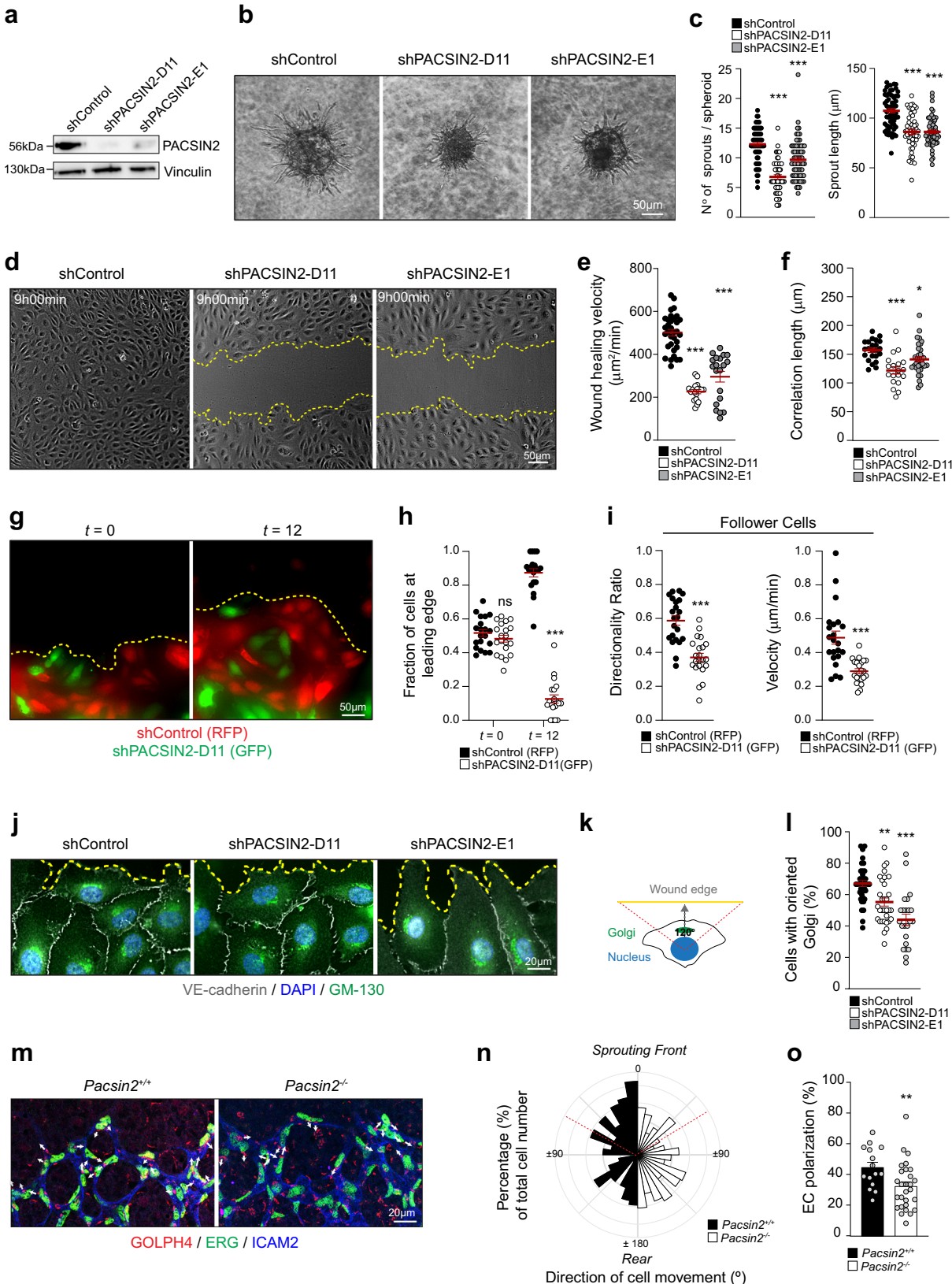

colocalization coefficients confirmed the strongest colocalization of PACSIN2 with EHD4 ($R = 0.62$) compared to EHD1 ($R = 0.38$), EHD2 ($R = 0.20$), and EHD3 ($R = 0.23$) at asymmetric AJs (Fig. 4c).

IF imaging further showed that EHD4, like PACSIN2[19], is recruited to the rear of asymmetric AJs in follower cells that are

aligned in the direction of collective migration (Supplementary Fig. 4). To study whether PACSIN2 and EHD4 bind to each other, we next conducted GFP-tag-based immunoprecipitations (IPs) on lysates of HUVECs expressing either PACSIN2-GFP or EHD4-GFP. Endogenous EHD4 was efficiently pulled down in PACSIN2-GFP IPs (Fig. 4d, e), and vice versa endogenous

**Fig. 2 PACSIN2 controls endothelial directed migration and angiogenic sprouting. a** Representative Western blot analysis of PACSIN2 and vinculin (loading control) expression in whole-cell lysates from HUVECs transduced with shControl, shPACSIN2-D11, and shPACSIN2-E1. Western blots were repeated three times with similar results. **b** Representative phase-contrast images of sprouting spheroids from HUVECs transduced with shControl, shPACSIN2-D11, and shPACSIN2-E1 after 16-h stimulation with VEGF. **c** Quantification of the number of sprouts per spheroid and the average sprout length of HUVECs transduced with shControl ($n = 63$ spheroids), shPACSIN2-D11 ($n = 52$ spheroids), and shPACSIN2-E1 ($n = 67$ spheroids). Data are from four independent experiments. $P < 0.0001$ when comparing shControl to shPACSIN2-D11 and when comparing shControl to shPACSIN2-E1. **d** Representative phase-contrast images of the scratch wound migration assay at 9 h post-scratch performed on HUVEC monolayers transduced with shControl, shPACSIN2-D11, and shPACSIN2-E1. The punctuated yellow lines indicate the boundaries of the wound. See Supplementary Movie 1 for time-lapse images of the scratch wound migration. **e** Quantification of wound-healing velocity measured in surface area per min of post-scratch HUVEC monolayers transduced with shControl ($n = 33$ movies), shPACSIN2-D11 ($n = 18$ movies), and shPACSIN2-E1 ($n = 20$ movies) from three independent experiments. $P < 0.0001$ when comparing shControl to shPACSIN2-D11 and when comparing shControl to shPACSIN2-E1. **f** Quantification of the correlation length of wound-healing time-lapse recordings of HUVECs transduced with shControl ($n = 23$ movies), shPACSIN2-D11 ($n = 19$ movies), and shPACSIN2-E1 ($n = 35$ movies) from at least four independent experiments using particle image velocimetry (PIV) analysis. $P < 0.0001$ when comparing shControl to shPACSIN2-D11 and $P = 0.0358$ when comparing shControl to shPACSIN2-E1. **g** Representative fluorescence images of HUVECs transduced with shControl-RFP or shPACSIN2 together with GFP in a scratch wound assay ($t = 0$ and $t = 12$ h after scratch). The punctuated yellow line highlights the migration front. **h** Quantification of the fraction of shControl (RFP) or shPACSIN2 (GFP) cells at the leading edge at $t = 0$ and $t = 12$ h after scratch. $P < 0.0001$ when comparing shControl to shPACSIN2 at $t = 12$ h using two-way ANOVA multiple comparisons. Data are from three independent experiments and 20 movies per condition. **i** Quantification of the average directionality and velocity of migrating follower shControl (RFP; $n = 22$ movies) or shPACSIN2 (GFP; $n = 22$ movies) cells. Directionality data were normally distributed and for comparison a paired two-tailed Student's $t$ test was performed. $P < 0.0001$ when comparing shControl to shPACSIN2-D11. The velocity data were not normally distributed, a paired two-tailed nonparametric Wilcoxon test was performed. $P < 0.0001$ when comparing shControl to shPACSIN2-D11. **j** Representative widefield IF images taken 5 h post-scratch of HUVEC monolayers transduced with shControl, shPACSIN2-D11, and shPACSIN2-E1 and stained for VE-cadherin (gray), DAPI (blue), and GM130 (green). The punctuated yellow lines indicate the boundaries of the wound. **k** A scheme to explain the quantification of Golgi orientation in migrating cells. Golgi was assessed as oriented if located within the 120° angle area directed toward the migration front of scratch wound assays. **l** Quantification of the percentage of cells with oriented Golgi in scratch wound assays with HUVECs transduced with shControl ($n = 39$ images), shPACSIN2-D11 ($n = 23$ images), and shPACSIN2-E1 ($n = 32$ images) from three independent experiments. $P = 0.0017$ when comparing shControl to shPACSIN2-D11 and $P < 0.0001$ when comparing shControl to shPACSIN2-E1. **m** Representative images of the sprouting front of retinas from control and $Pacsin2^{-/-}$ mouse littermates at P6 stained for ERG (green), ICAM2 (blue), and GOLPH4 (red). Arrows display the orientation of the Golgi in relation to the nucleus. **n** Rose plot of the polarization data distribution of quantified endothelial cells from control and $Pacsin2^{-/-}$ retinas at the first three rows of the vascular sprouting front. **o** Quantification of the percentage of polarized endothelial cells as defined within the ±60° range toward the sprouting front. $P = 0.0087$ when comparing $Pacsin2^{+/+}$ to $Pacsin2^{-/-}$ (two-sided Mann–Whitney test). All graphs represent mean ± SEM (error bars), and the statistical analysis was performed by a one-way ANOVA and Dunnett's multiple comparisons test unless stated otherwise. ns non-significant; $*P < 0.05$; $**P < 0.01$; $***P < 0.001$. Scale bars, 50 µm (**b, d, g**) and 20 µm (**j, m**). $t$ time. Source data are provided as a Source Data file.

PACSIN2 was co-immunoprecipitated in EHD4-GFP IPs (Fig. 4f, g). Together, these results indicate that PACSIN2 and EHD4 interact at the asymmetric AJs during collective migration.

Binding of PACSIN2 to EHD proteins occurs during the biogenesis of the recycling endosome—a tubular recycling compartment[56,57]. In addition, EHD4 has been shown to control cargo recycling[58]. The lipid-binding MICAL-like protein 1 (MICAL-L1) links EHD proteins to the endocytic recycling compartment, functions upstream of Rab proteins, and is a marker for endocytic recycling[59,60]. IF imaging of HUVECs revealed that MICAL-L1 is recruited to the trailing end of 31% of the asymmetric AJs (Fig. 4h, j). Next, to investigate whether MICAL-L1 is part of the tubular junctional PACSIN2-EHD4 complex at asymmetric AJs between HUVECs, we assessed colocalization of endogenous MICAL-L1 and ectopically expressed EHD4-GFP (Fig. 4i, k) or PACSIN2-GFP (Supplementary Fig. 5). IF-based imaging and Pearson's coefficient analyses revealed strong junctional colocalization of MICAL-L1 and EHD4-GFP ($R = 0.72$) (Fig. 4i, k), and MICAL-L1 and PACSIN2-GFP ($R = 0.82$) (Supplementary Fig. 5). These findings were further corroborated by the notion that endogenous MICAL-L1 was readily pulled down in PACSIN2-GFP IPs from HUVECs (Fig. 4d, l). Taken together, these results demonstrate that MICAL-L1 interacts with PACSIN2 and EHD4 at the trailing end of asymmetric AJs to form a tubular recycling compartment.

**The PACSIN2/EHD4/MICAL-L1 complex controls VE-cadherin trafficking at asymmetric AJs.** Asymmetric AJs are remodeled in a front–rear polarized fashion, and local PACSIN2 recruitment protects the integrity of VE-cadherin-based junctions[1,19]. Concordantly, the depletion of PACSIN2 leads to strongly augmented VE-cadherin internalization levels[19]. To investigate if the interaction of PACSIN2 with the EHD4/MICAL-L1 complex controls VE-cadherin-based junction turnover, we depleted EHD4 from HUVECs by two different *EHD4*-specific shRNAs. IF imaging revealed a clear loss of EHD4 from the trailing end of asymmetric AJs upon EHD4 depletion (Fig. 5a, b). Intriguingly, both the recruitment of PACSIN2 and MICAL-L1 to asymmetric AJs were also perturbed upon EHD4 depletion (Fig. 5c–f), indicating complete disruption of the junctional PACSIN2/EHD4/MICAL-L1 complex. Vice versa, depletion of PACSIN2 prevented the recruitment of EHD4 to asymmetric AJs (Supplementary Fig. 6). These experiments show that both EHD4 and PACSIN2 are needed to assemble the junctional PACSIN2/EHD4/MICAL-L1 complex.

HUVECs transduced with shEHD4 were still able to form VE-cadherin-based junctions and no major cytoskeletal changes were observed (Fig. 5 and Supplementary Fig. 7a). Fluorescence-activated cell sorting (FACS) analysis indicate that the depletion of EHD4 does not affect overall cell surface levels of VE-cadherin (Supplementary Fig. 7b). Next, we determined VE-cadherin intensity and localization in shControl, shPACSIN2, and shEHD4 monolayers by immunostaining the VE-cadherin molecules on the cell surface without permeabilization using an anti-VE-cadherin antibody that recognizes an extracellular epitope of VE-cadherin. In agreement with the FACS data and our earlier experiments with shPACSIN2 HUVECs[19], we observed no clear differences in VE-cadherin localization or intensity in the various knockdown conditions compared to control (Fig. 6a upper images). This

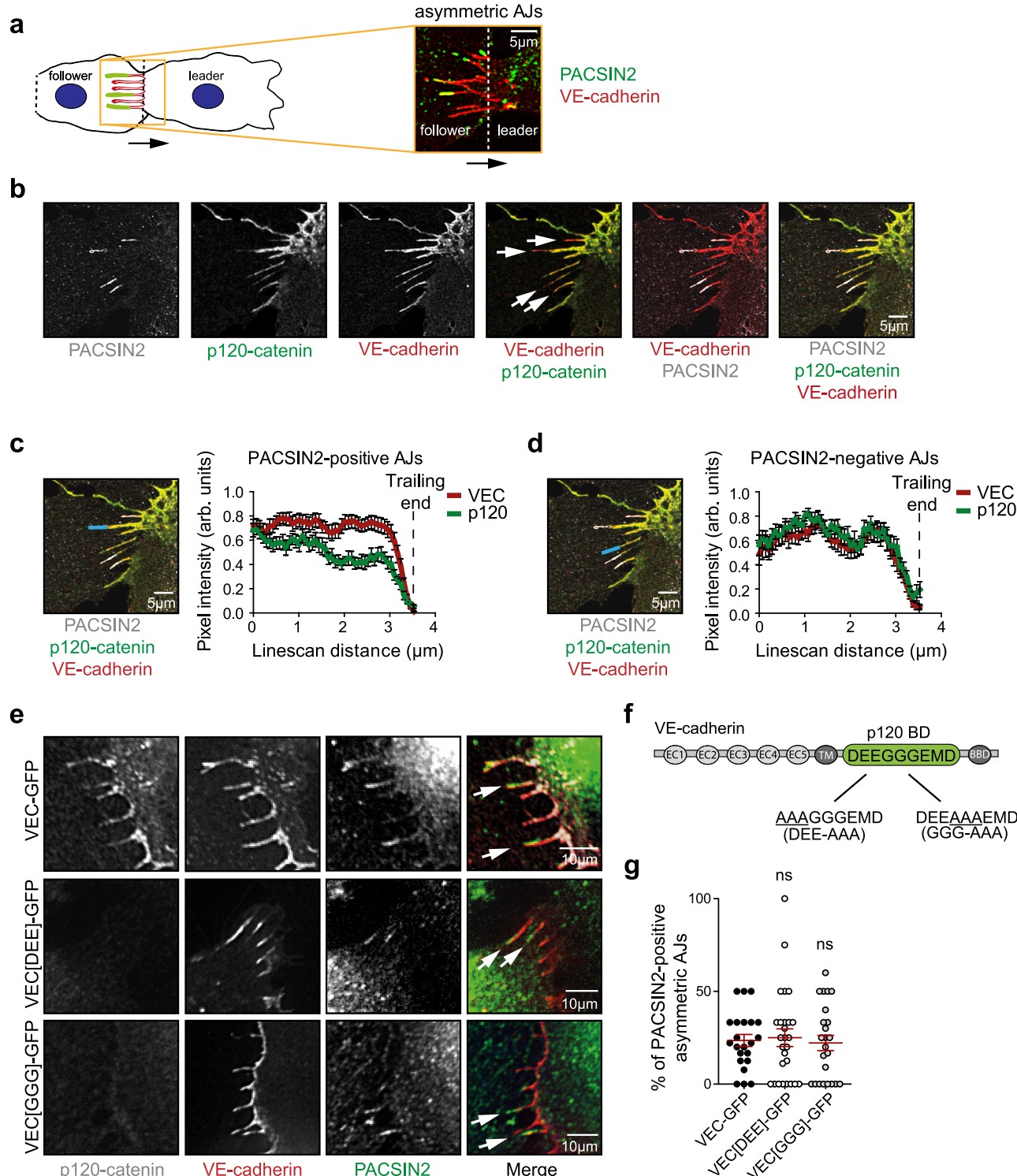

indicates that global VE-cadherin-based junction biogenesis pathways are unaffected. To study if EHD4, like PACSIN2, controls VE-cadherin trafficking, we pulse-labeled VE-cadherin molecules on the surface of shControl, shPACSIN2, and shEHD4 HUVECs and followed their turnover and internalization. These experiments show that in a time course of 2 h, a larger proportion of VE-cadherin is internalized and accumulating in intracellular vesicles upon depletion of PACSIN2 or EHD4 (Fig. 6a, b).

Live imaging of HUVECs expressing EHD4-GFP and VE-cadherin-mCherry showed that the recruitment of EHD4 relates to rapid movements of the asymmetric AJs (Fig. 6c and

Supplementary Movie 3). To investigate whether EHD4 controls local turnover of the AJs, VE-cadherin-GFP was expressed in shControl and shEHD4 HUVECs. Subsequent live cell imaging experiments clearly showed that the asymmetric AJs are gradually turning over during their movement through subtle internalization events (Fig. 6d and Supplementary Movie 4). Importantly, upon the depletion of EHD4, the asymmetric AJs are not turning over in a smooth fashion, but instead elongate until they break (Fig. 6d and Supplementary Movie 4). This junctional defect in shEHD4 HUVECs likely underlies the accumulation of VE-cadherin-positive vesicles in the pulse-chase experiments.

**Fig. 3 PACSIN2 recruitment to asymmetric AJs occurs in parallel with the dissociation of p120-catenin from VE-cadherin. a** A schematic representation and an IF image zoomed-in on asymmetric AJs at the junctional interface of collectively migrating HUVECs. Stained PACSIN2 is represented in green and VE-cadherin in red. **b** IF images of asymmetric AJs between HUVECs stained for PACSIN2 (white), p120-catenin (green), and VE-cadherin (red) and dual and triple channel merges. **c** On the left—a representative site of a linescan analysis of VE-cadherin and p120-catenin along a PACSIN2-positive asymmetric AJ (indicated with a solid blue line of 3.5-μm length). On the right—a graph with averages of VE-cadherin and p120-catenin intensities, normalized and corrected for background, measured along PACSIN2-positive asymmetric AJs ($n = 21$ asymmetric AJs) from three independent experiments. **d** On the left—a representative site of a linescan analysis of VE-cadherin and p120-catenin along a PACSIN2-negative asymmetric AJ. On the right—a graph with averages of VE-cadherin and p120-catenin intensities, normalized and corrected for background, measured along PACSIN2-negative asymmetric AJs ($n = 13$ asymmetric AJs) from three independent experiments. **e** Representative widefield IF images of asymmetric AJs in HUVECs depleted for endogenous VE-cadherin and rescued with ectopic expression of VE-cadherin-GFP, VE-cadherin[GGG]-GFP, or VE-cadherin[DEE]-GFP (red) and stained for endogenous p120-catenin (white) and PACSIN2 (green). The white arrows point to PACSIN2-positive asymmetric AJs. Corresponding Western blot analysis of endogenous VE-cadherin depletion is shown in Supplementary Fig. 3b. **f** A schematic representation of the p120-catenin-binding domain mutated VE-cadherin variants. **g** Quantification of the percentage of PACSIN2-positive asymmetric AJs detected in VE-cadherin-GFP ($n = 22$ images), VE-cadherin[DEE]-GFP ($n = 27$ images), VE-cadherin[GGG]-GFP ($n = 24$ images) expressing HUVECs from three independent experiments. The statistical analysis was performed by a one-way ANOVA and Dunnett's multiple comparisons test. ns non-significant. All graphs represent mean ± SEM (error bars). Scale bars, 5 μm (**a–d**) and 10 μm (**e**). AJs adherens junctions, Arb. units arbitrary units, VEC VE-cadherin. Source data are provided as a Source Data file.

We next investigated if EHD4 controls remodeling of AJs during endothelial collective migration. To study this, we live-labeled VE-cadherin in shControl or shEHD4 HUVEC monolayers and induced a scratch wound to trigger collective migration. Both shEHD4 and shControl HUVECs established asymmetric AJs between the leader and follower cells (Fig. 6e). Live imaging revealed that the asymmetric AJs in shControl HUVECs were gradually resolved, allowing for the dynamic breaking and making of cell–cell contacts to support the coordination of polarized cell migration (Fig. 6e and Supplementary Movie 5). Strikingly, in the shEHD4 HUVECs the majority of asymmetric AJs persisted longer, and the asymmetric AJs elongated, supporting the notion that VE-cadherin is not properly trafficked (Fig. 6e and Supplementary Movie 5). This resulted in increased maximal junctional length and prolonged lifetime of the asymmetric AJs between EHD4 knockdown cells (Fig. 6f, g). Together, these findings clearly show that the PACSIN2/EHD4/MICAL-L1 complex mediates proper VE-cadherin trafficking at asymmetric junctions.

**The PACSIN2/EHD4/MICAL-L1 complex is needed for endothelial collective migration and sprouting angiogenesis.** Next, we validated the functional importance of the PACSIN2/EHD4/MICAL-L1 complex for endothelial collective behavior. HUVECs depleted of EHD4 exhibited an approximate twofold delay in wound closure compared to control cells (Fig. 7a, b and Supplementary Movie 6), similar to the migration capacity of shPACSIN2 HUVECs (Fig. 2d, e). The correlation length between neighboring migrating shEHD4 HUVECs is reduced compared to shControl (Fig. 7c), indicative of a loss of collectively coordinated endothelial cell migration upon EHD4 depletion. In addition, the depletion of EHD4 impaired VEGF-induced sprouting and sprout elongation in 3D collagen matrices (Fig. 7d, e). Based on the notion that these effects are comparable to the functional defects in shPACSIN2 HUVECs (Fig. 2), we conclude that assembly of the junctional PACSIN2/EHD4 complex supports asymmetric AJ remodeling for collective endothelial migration.

We next investigated the role of EHD4 in retinal angiogenesis. Immunostaining of P6 wild-type retinas showed that EHD4 is highly expressed in the retinal vasculature and associates with VE-cadherin junctions to some extent (Supplementary Fig. 8a). To establish the importance of EHD4 in vascular development, we generated homozygous $Ehd4^{-/-}$ knockout mice through homologous recombination and ubiquitous Cre-loxP recombination (Supplementary Fig. 8b, see "Methods"). Depletion of EHD4 was confirmed in the retina and lung tissue of the $Ehd4^{-/-}$ mice,

and $Ehd4$ gene deletion did not affect the expression of the related EHD1, EHD2, and EHD3 isoforms (Supplementary Fig. 8c). $Ehd4^{-/-}$ mice are viable, fertile, and develop toward adulthood, including development of retinal vasculature with equal number of vascular branch points, sprouts, and endothelial cell numbers (Fig. 7f, g). Notably, the deletion of EHD4 resulted in a decrease in the endothelial sprout length and an increase in the number of endothelial cell clusters at the angiogenic front of the developing vasculature in P6 retinas (Fig. 7h, i). Also, we observed a slight increase (not significant) in cytoplasmic VE-cadherin signal in the angiogenic endothelial cells in $Ehd4^{-/-}$ retinal vasculature compared to controls (Fig. 7 j, k).

Taken together, these results show that EHD4, like PACSIN2, controls endothelial collective migration and angiogenic sprouting. Altogether, these data support the notion that the PACSIN2/EHD4/MICAL-L1 complex forms a tubular trafficking compartment at asymmetric AJs that is required for coordinated endothelial cell migration and rearrangements during sprouting angiogenesis.

## Discussion

The ability of the endothelium to adapt as collective tissue is crucial for vascular development and maintenance of the vascular barrier in the mature organism. Communication between leading tip cells and following stalk cells orchestrates the development of a proper vascular bed and involves direct anchoring, signaling, and orientation between the endothelial cells[2,7]. AJs are essential hubs of cell-to-cell communication during collective tissue behavior. Although the importance of mechanical coupling between the leader and follower cells through cadherin-based adhesions has been well established[11,61–64], we have only just begun to appreciate the extent to which junctional-derived signaling events guides collective behavior. Our data reveal that the F-BAR protein PACSIN2 acts as a sensor between leader and follower cells by recruiting the trafficking modulators EHD4 and MICAL-L1 to the rear of asymmetric AJs. We find that formation of the junctional PACSIN2/EHD4/MICAL-L1 complex drives proper turnover of VE-cadherin at remodeling AJs (Fig. 8), and we provide proof for the importance of PACSIN2/EHD4/MICAL-L1 complex signaling for endothelial collective migration and sprouting angiogenesis in vitro and in vivo. Together, these results put forward asymmetric AJs as important players at the endothelial leader and follower cell interface, in addition to the well-established angiogenic signaling cascades, such as VEGF-VEGFR and Delta-Notch signaling that shape the angiogenic front.

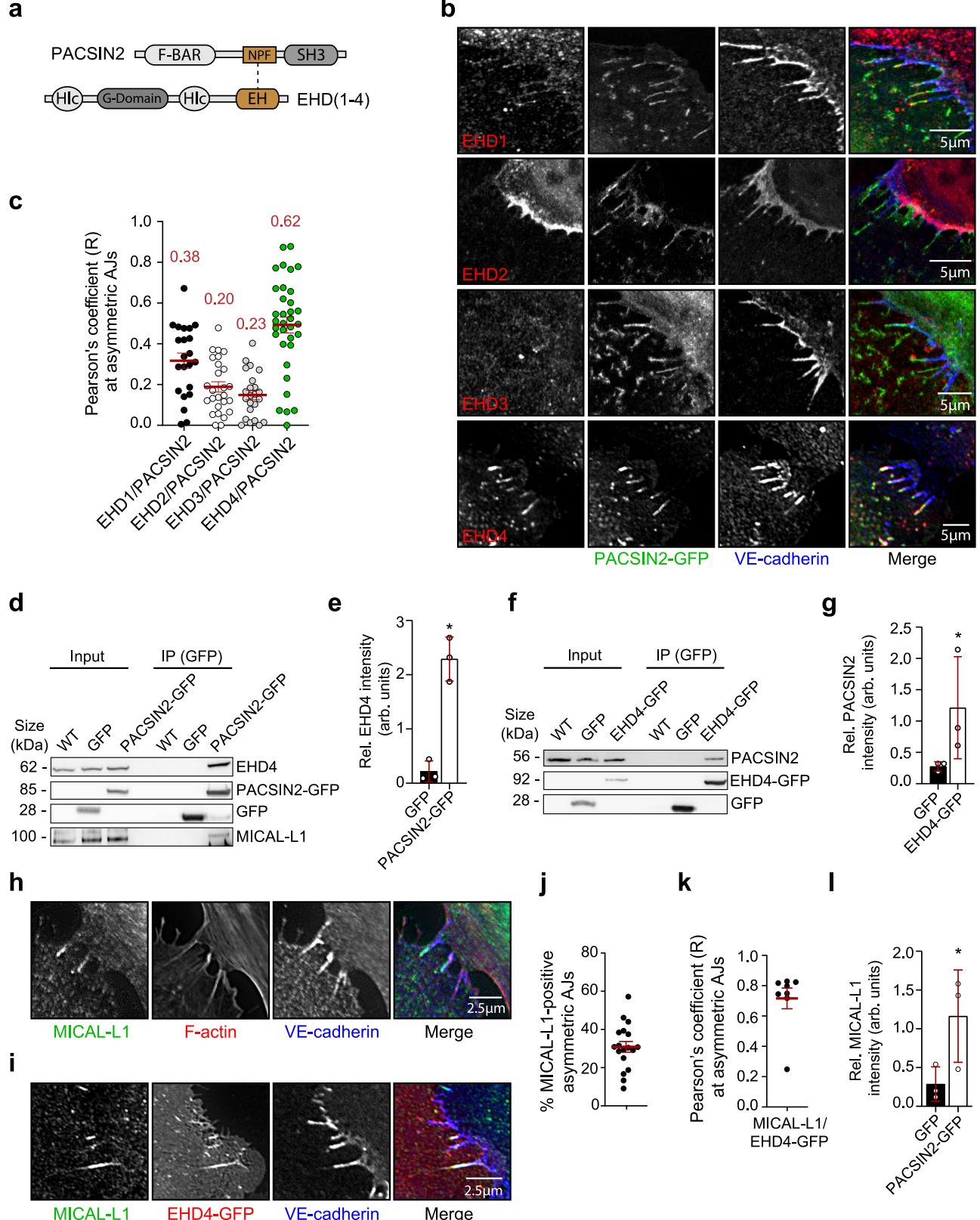

Collective migration requires polarized remodeling of the AJs[2,18,65], which is mediated by internalization, sorting, recycling, and (re)formation of cadherin-based adhesions[43,46,50,66–68]. Although the presence of asymmetric AJs has been numerously described[3,19,31,69,70], their role and importance in endothelial migration and vascular development remained to be established.

Our data reveal that PACSIN2 is recruited to asymmetric AJs that are primed for endocytosis by the dissociation of p120-catenin from the VE-cadherin complex. At the trailing rear of asymmetric AJs, PACSIN2 binds to the recycling regulators EHD4 and MICAL-L1. Our results further implicate the importance of this junctional tubular trafficking compartment in balancing the

**Fig. 4 PACSIN2, EHD4, and MICAL-L1 interact at asymmetric AJs. a** A schematic representation of the domain structures of PACSIN2 and EHD proteins. NPF Asn-Pro-Phe motif; SH3 SRC homology 3 domain; Hlc helical domain. **b** Confocal and widefield IF images of asymmetric AJs in HUVECs expressing PACSIN2-GFP (green) and stained for EHD1, EHD2, EHD3, or EHD4 (red) and VE-cadherin (blue). **c** Pearson's correlation analysis of PACSIN2 and EHD isoform fluorescent signals at the asymmetric AJs. The red numbers represent the mean Pearson's correlation values (R). Data are derived from three independent experiments; EHD1/PACSIN2 ($n = 22$ asymmetric AJs), EHD2/PACSIN2 ($n = 26$ asymmetric AJs), EHD3/PACSIN2 ($n = 23$ asymmetric AJs), EHD4/PACSIN2 ($n = 34$ asymmetric AJs). **d** Representative Western blot analysis for EHD4, GFP, and MICAL-L1 on whole-cell lysates and GFP pull-down samples derived from non-transduced HUVECs (WT), HUVECs expressing GFP, or PACSIN2-GFP. **e** Quantification of relative EHD4 intensity detected in GFP pull-down samples from HUVECs expressing GFP or PACSIN2-GFP ($n = 3$ independent experiments). The signal is normalized to the amount of protein in the whole-cell lysate. $P = 0.0311$ when comparing GFP to PACSIN2-GFP. **f** Representative Western blot analysis for PACSIN2, EHD4, and GFP on whole-cell lysates and GFP pull-down samples derived from non-transduced HUVECs (WT), HUVECs expressing GFP or EHD4-GFP. **g** Quantification of relative PACSIN2 intensity detected in GFP pull-down samples from HUVECs expressing GFP or EHD4-GFP ($n = 3$ independent experiments). The signal is normalized to the amount of protein in the whole-cell lysate. $P = 0.0375$ when comparing GFP to EHD4-GFP. **h** Widefield IF images of asymmetric AJs in HUVECs stained for MICAL-L1 (green), F-actin (red), and VE-cadherin (blue). **i** Widefield IF images of asymmetric AJs in HUVECs expressing EHD4-GFP (red) stained for MICAL-L1 (green) and VE-cadherin (blue). **j** Quantification of the percentage of MICAL-L1-positive asymmetric AJs in HUVECs ($n = 18$ images from three independent experiments). **k** Pearson's correlation analysis of fluorescent signal of MICAL-L1 and EHD4-GFP at the asymmetric AJs ($n = 8$ images from three independent experiments). **l** Quantification of relative MICAL-L1 intensity detected in GFP pull-down samples from HUVECs expressing GFP or PACSIN2-GFP ($n = 3$ independent experiments). The signal is normalized to the amount of protein in the whole-cell lysate. $P = 0.0436$ when comparing GFP to PACSIN2-GFP. All graphs represent mean ± SEM (error bars), and the statistical analysis was performed by a paired two-tailed $t$ test. *$P < 0.05$. Scale bars, 5 μm (**b**) and 2.5 μm (**h, i**). IP immunoprecipitation, Arb. units arbitrary units. Source data are provided as a Source Data file.

breaking and making of AJs during endothelial collective behavior. Recent experiments using transgenic *CDH5* knock-in mouse models, established that the p120-mediated turnover of VE-cadherin is important for endothelial collective migration and endothelial cell polarization[51]. Our current finding that the PACSIN2-EHD4 complex controls VE-cadherin turnover and drives endothelial collective migration and angiogenesis is in strong agreement with the notion that proper trafficking of VE-cadherin is needed for endothelial polarity and retinal angiogenesis.

EHD proteins bind to the NPF motifs of PACSIN2 and their association has been observed in cilia of fibroblasts and epithelial cells, as well as in the soma and neurites of neuronal cells[57,71]. Also, an interaction between PACSIN2, EHD1, and MICAL-L1 was previously shown during the formation of the recycling endosome at the nuclear periphery[56], where it receives cargo sorted for recycling from the early endosome[55,72]. Our results reveal that the PACSIN2/EHD4/MICAL-L1 complex is locally generated at asymmetric AJs to fit the need of the endothelium to direct AJ turnover in collective migration. We observed that MICAL-L1 is also recruited to a fraction of remodeling AJs that are negative for PACSIN2 (Supplementary Fig 5). This opens up the possibility that additional NPF-containing BAR proteins and/or EHD isoforms are involved in the turnover of AJs of different tubular sizes and/or shapes.

The PACSIN2/EHD4/MICAL-L1 complex may contribute to VE-cadherin trafficking by targeting the junctional membrane. EHD1 is known to actively sculpt membranes, thereby driving membrane scission and promoting endocytic recycling[73]. Likewise, expression of PACSIN2, EHD4, and EHD1 has been shown to contribute to infectious virus spread by controlling membrane fission in host cells[74,75]. Alternatively, the PACSIN2/EHD4/MICAL-L1 complex may tighten the connections between endocytic pathways and cytoskeletal adaptations that provide polarized cues across AJs for collectively migrating cells[26,40,70].

Here, we have focussed on the primary signaling events at the front–rear interface between migrating neighboring cells, where pulling from the leader cells leads to force-dependent remodeling of the AJs. Interestingly, similar junction remodeling is observed in response to other mechanical triggers such as shear stress[76] and monolayer stiffness heterogeneity[77,78]. We therefore expect that the formation of the PACSIN2/EHD4/MICAL-L1 complex is not limited to collective cell migration during angiogenesis and wound healing. We surmise that the complex forms as an

adaptation to a variety of mechanical stimuli that promote rapid turnover of cell–cell contacts and junctional membrane tension to protect junction stability and vascular integrity. Given the notion that PACSIN2 is an ubiquitously expressed protein, experimental approaches using endothelial-specific conditional knockout models might further reveal the function of the PACSIN2/EHD4/MICAL-L1 complex as mechanosensing module and its involvement in vascular homeostasis and pathology.

## Methods

**Cell culture**. We used primary HUVEC (up to passage 5) pooled from different donors (obtained under informed consent) from Lonza (Cat # C2519A). HUVEC were cultured on gelatin-coated cell culture dishes in Endothelial Cell Growth Medium 2 supplemented with the Growth Medium 2 Supplement Pack (EGM-2) from Promocell. HEK293T cells (ATCC, CRL-3216) were cultured in Dulbecco's Medium Eagle medium with L-glutamine supplemented with 10% FCS and penicillin (100 units mL$^{-1}$), and streptomycin (100 mg mL$^{-1}$) (ThermoFisher). All cells were cultured at 37 °C and 5% $CO_2$.

**Mice**. We used homologous recombination and the Cre-*loxP* recombination to create mice with a *loxP*-flanked (floxed) expression of the *Pacsin2* gene. The targeting vector contained a single *loxP* site in intron 3 and an FRT-site-flanked neo cassette with an additional *loxP* site in intron 4, to allow conditional removal of exon 4 (Supplementary Fig. 1b). Two independent transfected ES clones with the correct recombination event were identified by Southern blotting and injected into blastocysts. Mice carrying the *Pacsin2*$^{flox(neo)}$ allele were then bred to homozygosity and crossed with transgenic Flp-recombinase deleter mice to remove the FRT-site flanked neo cassette (Supplementary Fig. 1b). The resulting homozygous *Pacsin2*$^{flox}$ mice appeared normal, indicating that the genetic manipulation had not altered the function of PACSIN2. To generate animals lacking PACSIN2 expression, *Pacsin2*$^{flox}$ mice were bred to transgenic Cre-deleter mice to produce PACSIN2-deficient animals carrying the *Pacsin2*$^{null}$ allele. The removal of exon 4 was confirmed by Southern blotting on DNA from the progeny (Supplementary Fig. 1c). To check for homologous recombination, ScaI-digested genomic DNA was analyzed by Southern blotting using a 1.3-kb XhoI-BamHI DNA fragment of a genomic subclone as external probe (corresponding to nt 165762–166609 of acc.no. AL583889). This probe identifies an 8.8-kb fragment and a 6.4-kb fragment in the wild-type and mutant alleles, respectively. We backcrossed these mice onto the C57BL/6 background for six generations. In homozygous PACSIN2-deficient mice, very low levels of *Pacsin2* transcripts could be detected by Northern blotting (Supplementary Fig. 1d), but tested retina and lung tissues lacked the PACSIN2 protein (Supplementary Fig. 1e). For Northern blot analysis, RNA was isolated from freshly prepared tissues using guanidinium thiocyanate lysis, and poly(A) + RNA was enriched by using the Oligotex mRNA kit (Qiagen). After fractionation of the RNA by a 1% agarose-formaldehyde gel, and subsequent capillary transfer onto Hybond XL membrane (Amersham Pharmacia Biotech), hybridization was performed in a formamide mix and radiolabelled probes were generated by using a labeling kit (TaKaRa). For isoform-specific hybridization[79], the following eluted DNA fragments of the corresponding *Pacsin* cDNA clones were used as probes: a 0.36-kb SmaI fragment corresponding to nt 704–1067 of *Pacsin1* cDNA (acc.no. X85124), a 0.25-kb SmaI/NheI fragment corresponding to nt 718–984 of *Pacsin2*

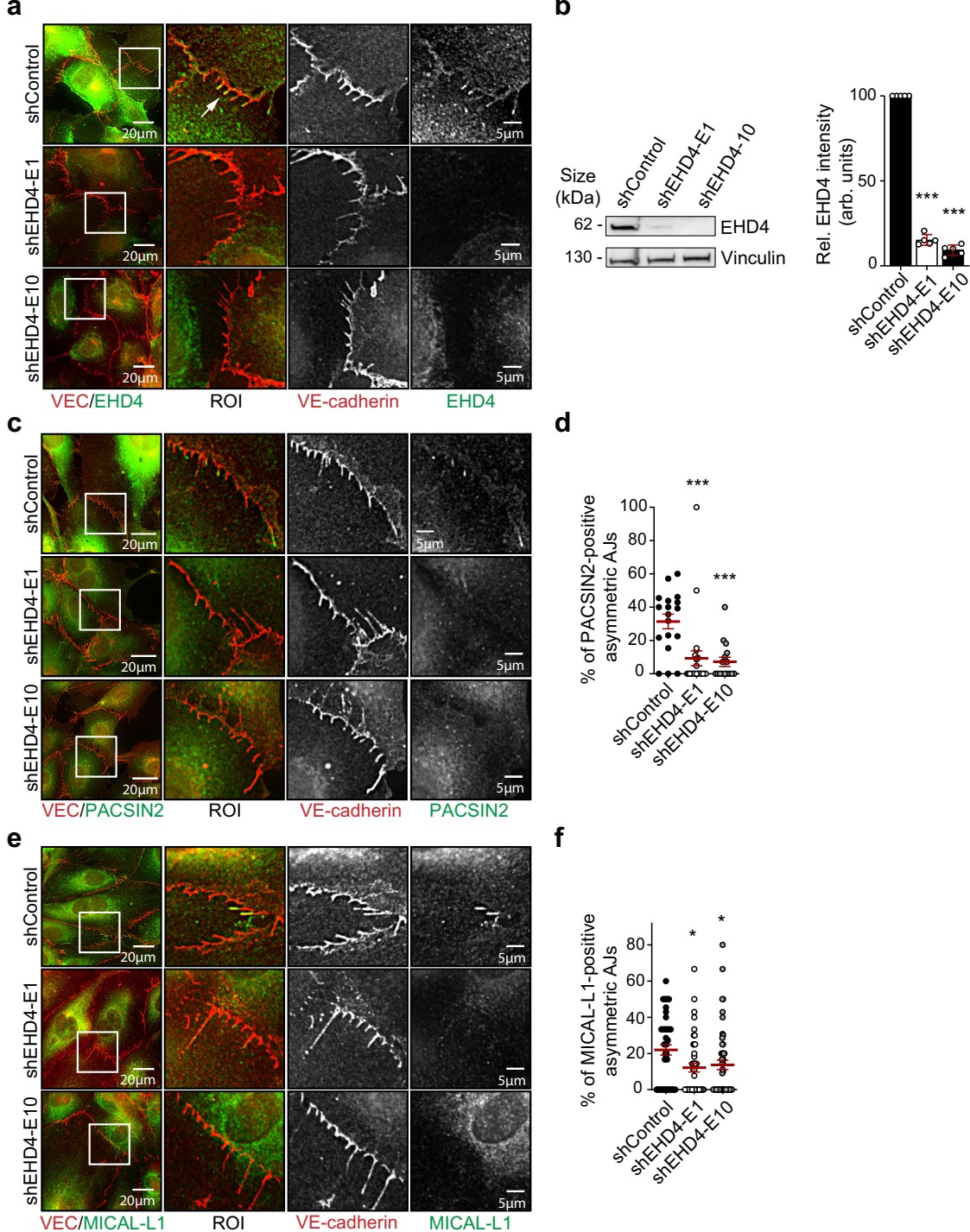

cDNA (acc.no. XM_030248566), and a 0.52-kb SmaI fragment corresponding to nt 854–1379 of *Pacsin3* cDNA (acc. no. NM_028733), respectively. The filters were stringently washed before autoradiography. Prior ethidium bromide staining of the gel and hybridization with glycerin aldehyde-3-phosphate dehydrogenase cDNA were used to control equal loading and to check RNA integrity.

The *Ehd4* mutant mice were commercially generated by inGenious Targeting Laboratory, Inc. (Stony Brook). Briefly, for the construction of the floxed *Ehd4* targeting vector, a genomic fragment spanning a total of 10.05 kb of *Ehd4* sequence including exon 1 was isolated from a C57BL/6 BAC clone (RP23: 108D15) using a homologous recombination-based technique. The final construct was designed such that the long homology arm extends 6.8 kb from the 5′-end of the single *loxP* site located upstream of exon 1 (Supplementary Fig. 8b). The short homology arm extends 2.2 kb from the 3′-end of the *loxP*/FRT-site-flanked neo cassette, which was inserted downstream of exon 1. The target region spans ~1.0 kb containing exon 1. The resulting targeting construct was linearized by cutting at the NotI restriction

site located within the vector backbone 3′ of the sequence homologous to *Ehd4*. Linearized DNA was electroporated into C57BL6/129SvEv hybrid embryonic stem cells. Homologous recombination events were identified by Southern blot analysis using external and internal probes. Four correctly targeted ES cell clones were isolated and injected into C57/BL6 blastocysts to generate chimeric mice. To generate EHD4-deficient animals, homozygous *Ehd4*flox mice were crossed to Cre-deleter mice, yielding ubiquitous disruption of the *Ehd4* gene (Supplementary Fig. 8b).

All mouse studies were performed after ethical approval for breeding and maintaining mice by local government authorities (Landesamt für Natur, Umwelt und Verbraucherschutz Nordrhein-Westfalen (permission number no. 81-02.04.2019A215, Germany) in accordance with the German animal protection law and with permission to sacrifice mice for scientific purpose. Animals were housed in the Center for Molecular Medicine animal care facility of the University of Cologne under standard pathogen-free, temperature- and humidity-controlled

**Fig. 5 EHD4 is needed to form a junctional PACSIN2/EHD4/MICAL-L1 complex at asymmetric AJs. a** Representative widefield images of asymmetric AJs in HUVECs transduced with shControl, shEHD4-E1, and shEHD4-E10 stained for VE-cadherin (red) and EHD4 (green). The white squares indicate the ROIs that are magnified in the right panels. The white arrow points to EHD4-positive asymmetric AJs. Imaging experiments were repeated three times with similar results. **b** Representative Western blot analysis of EHD4 and vinculin (loading control) protein levels in whole-cell lysates from HUVECs transduced with shControl, shEHD4-E1, or shEHD4-E10. The graph represents quantification of EHD4 protein levels in lysates of shEHD4 transduced HUVECs, normalized to the levels of the protein in lysates from HUVECs transduced with shControl ($n = 5$ independent experiments). $P < 0.0001$ when comparing shControl to shEHD4-E1 and when comparing shControl to shEHD4-E10. **c** Representative widefield images of asymmetric AJs in HUVECs transduced with shControl, shEHD4-E1, and shEHD4-E10 stained for VE-cadherin (red) and PACSIN2 (green). The white squares indicate the ROIs that are magnified in the right panels. **d** Quantification of the percentage of PACSIN2-positive asymmetric AJs in HUVECs transduced with shControl ($n = 18$ images), shEHD4-E1 ($n = 24$ images) or shEHD4-E10 ($n = 16$ images) from three independent experiments. $P = 0.007$ when comparing shControl to shEHD4-E1 and $P = 0.008$ when comparing shControl to shEHD4-E10. **e** Representative widefield images of asymmetric AJs in HUVECs transduced with shControl, shEHD4-E1, and shEHD4-E10 stained for VE-cadherin (red) and MICAL-L1 (green). The white squares indicate the ROIs that are magnified in the right panels. **f** Quantification of the percentage of MICAL-L1-positive asymmetric AJs in HUVECs transduced with shControl ($n = 46$ images), shEHD4-E1 ($n = 44$ images), or shEHD4-E10 ($n = 51$ images) from three independent experiments. $P = 0.022$ when comparing shControl to shEHD4-E1 and $P = 0.0493$ when comparing shControl to shEHD4-E10. The graphs represent mean ± SEM (error bars). The statistical analyses were performed by a one-way ANOVA and Dunnett's multiple comparisons test. *$P < 0.05$, ***$P < 0.001$. Scale bars—20 and 5 μm (**a**, **c**, **e**). ROI region of interest, Arb. units arbitrary units. Source data are provided as a Source Data file.

conditions with a 12-h light/dark schedule and provided with food and water ad libitum. Experimental procedures were performed according to all relevant ethical regulations for animal testing and research.

**Antibodies**. Detection of PACSIN2 was performed with purified rabbit polyclonal anti-human PACSIN2 antibody (Cat # AP8088b; diluted 1/100 for IF; 1/5000 for Western Blot) obtained from Abgent and affinity-purified antibodies against mouse PACSIN2 and PACSIN3 (dilution 1/100 for IF) raised in rabbit were from the laboratory of M.P. We used purified mouse anti-human p120-catenin antibody (Clone 98/pp120, Cat # 610134; diluted 1/100 for IF) obtained from BD Biosciences. Imaging of VE-cadherin was performed with the following antibodies: purified goat anti-human VE-cadherin (Clone C-19, Cat # SC-6458, diluted 1/100) from Santa Cruz, rabbit polyclonal anti-VE-cadherin (Cat # 160840, diluted 1/100 for IF) from Cayman Chemical, mouse anti-cadherin-5 (Clone 75, Cat # 610252, diluted 1/100 for IF; and 1/1000 for Western Blot), rat anti-mouse VE-cadherin (BD Bioscience, #555289, diluted 1:50 for IF retinas), and directly labeled Alexa Fluor-647 mouse anti-human CD144 (Clone 55-7H1; Cat # 561567; diluted 1/200) from BD Biosciences. Affinity-purified antibodies against human EHD1, EHD2, EHD3, and EHD4 (dilution 1/100 for IF, 1/1000 for Western Blot) raised in rabbit were a gift from the laboratory of M.P. We used rabbit monoclonal anti-ERG (Clone EPR3864, #AB92513, diluted 1:400 for IF) from Abcam, polyclonal goat anti-endocan/ESM1 (1/100 for IF, Cat # AF1999) from R&D Systems, purified rat anti-ICAM2 (Clone 3C4 (mIC2/4), 1/100 for IF, Cat # 553326) from BD Pharmingen, and purified polyclonal rabbit anti-human MICAL-L1 (Cat # NBP2-55389; diluted 1/100 for IF, 1/1000 for Western Blot) antibody from Novus Biologicals. IF of Golgi was performed with purified mouse anti-GM130 (clone 35, Cat # 610823, diluted 1/200) antibody obtained from BD Biosciences in HUVECs and with the rabbit polyclonal anti-GOLPH4 (1/100 for IF, Cat # ab28049) from Abcam in mouse retinas. IF of cell nuclei was performed with DAPI (Invitrogen; diluted 1/1000). To perform IF on F-actin, we used PromoFluor-415 Phalloidin (Cat # PK-PF415-7-01, diluted 1/200) from Promokine. For IF with fixed cells, the following secondary antibodies from Invitrogen (diluted 1/200) were used: chicken anti-mouse Alexa Fluor-488 (Cat # A21200), chicken anti-rabbit Alexa Fluor-488 (Cat # A21441), chicken anti-mouse Alexa Fluor-594 (Cat # A21201), chicken anti-rabbit Alexa Fluor-594 (Cat # A21442), chicken anti-mouse Alexa Fluor-647 (Cat # A21463), chicken anti-rabbit Alexa Fluor-647 (Cat # A21443), and chicken anti-goat Alexa Fluor-647 (Cat # A21469). For IF stainings in the retina, the following labels and secondary antibodies from Invitrogen were used at 1/300 dilution: Isolectin GS-IB4 Alexa Fluor-488 (Cat # I21411), Isolectin GS-IB4 Alexa Fluor-568 (Cat # I21412), Isolectin GS-IB4 Alexa Fluor-647 (Cat # I32450), goat anti-rabbit Alexa Fluor-488 (Cat # A11008), goat anti-rabbit Alexa Fluor-568 (Cat # A11011), goat anti-rat Alexa Fluor-488 (Cat # A11006), and goat anti-rat Alexa Fluor-633 (Cat # A21094). As loading control for Western Blot, we used mouse monoclonal anti-human vinculin (Clone hVIN-1, Cat # V9131, diluted 1/1000) antibody from Sigma-Aldrich. To detect GFP in Western Blot, we used mouse anti-GFP (Clone B-2, Cat # sc-9996, dilution 1/1000) antibody from Santa Cruz. For Western Blot protein detection, we used horseradish peroxidase-coupled goat anti-mouse (A28177) and goat anti-rabbit (A27036) secondary antibodies (diluted 1/1000) from Invitrogen.

**DNA plasmids and lentiviral transduction**. PACSIN2 and EHD4 knockdowns were performed using pLKO.1 lentiviral vectors expressing shRNAs targeting human *PACSIN2* or *EHD4*. The shRNA constructs were obtained from The RNAi Consortium (TRC) library[80]. For PACSIN2 knockdown, we used MISSION TRC1

clones 0000037980 and 0000037983 designated as clone shPACSIN2-E1 and D11 throughout the manuscript. For EHD4 knockdown, we used MISSION TRC1 clones 0000053400 and 0000053401, designated as shEHD4-E10 and shEHD4-E1 throughout the manuscript. As a negative control, we used non-targeting shRNA (SHC002) from Sigma-Aldrich. shVE-cadherin-3′UTR was created by allowing two oligos containing the shRNA sequence 5′-CCGGTGGATAGCAAACTCCAGGT TCCCTCGAGGGGAACCTGGAGTTTGCTATCCTTTTTG-3′ to self-ligate. The product was then inserted into a modified version of pLKO.1 EV U6 backbone between the AgeI and EcoRI restriction sites. For ectopic expression of EHD4-GFP, human EHD4 cDNA was amplified by PCR from a peGFP-EHD4 plasmid (gift from Prof. Dr Steve Caplan). The PCR product was then cloned into a pLV-CMV-ires-puro vector using the Sequence- and Ligation-Independent Cloning method[81] and XbaI and NheI restriction sites. The Aspartate-Glutamate-Glutamate (GACG AGGAG) to Alanine-Alanine-Alanine (GCAGCAGCA) mutations at position 646–648 and Glycine-Glycine-Glycine (GGCGGCGGC) to Alanine-Alanine-Alanine (GCAGCAGCA) mutations at position 649–651 in the VE-cadherin cytoplasmic domain were achieved by site-directed mutagenesis in the peGFP-VE-cadherin plasmid[15]. Next, these plasmids were digested with PstI and Bpu1102I restriction enzymes. The 816-bp fragment, containing the mutations, was exchanged with the wild-type fragment from lentiviral plasmid pLV-CMV-VE-cadherin-eGFP-ires-puro[15] to generate the lentiviral pLV-CMV-VE-cadherin-[DEE646-648AAA]-eGFP and pLV-CMV-VE-cadherin-[GGG649-651AAA]-eGFP plasmids. The lentiviral expression constructs pLV-PACSIN2-GFP, pLV-VE-cadherin-mCherry, and pLV-p120-catenin-mCherry have been described before[15,19]. Lentiviral particles were produced in HEK293T cells, which were transiently transfected with third-generation packaging constructs and the lentiviral expression vector of interest using Trans-IT LTI (Mirus). HUVECs at 60% confluency were transduced with the lentiviral particles overnight. HUVECs transduced with shRNAs were analyzed at least 72 h post transduction. For primer sequences, see Supplementary Table 1.

**IF staining**. For standard IF stainings, HUVECs were cultured on coverslips coated with 5-μg ml$^{-1}$ human plasma fibronectin (Sigma-Aldrich). Cells were then fixed in 4% PFA diluted in PBS supplemented with 1-mM $CaCl_2$ and 0.5-mM $MgCl_2$ (PBS++) for 10 min. After fixation, the cells were permeabilized with 0.5% Triton X-100 in PBS and blocked with 2% bovine serum albumin (BSA) in PBS. Primary and secondary antibodies were diluted in 0.5% BSA in PBS and incubated in dark for 1 h each. After each incubation, the coverslips were thoroughly washed in 0.5% BSA in PBS. After the last wash, the coverslips were mounted on microscope slides in Mowiol4-88 (Calbiochem, #475904) and DABCO (Sigma-Aldrich, D27802) solution.

For retinal IF, isolated eyes from postnatal day 6 mice were fixed in 4% PFA in PBS for 1 h on ice and washed in PBS for at least 10 min. Retinas were dissected from the eyes and fixed with 4% PFA in PBS for 1 h on ice. Retinas were washed with PBS and blocked with blocking buffer (1% BSA, 0.3% Triton X-100 in PBS) overnight at 4 °C. Next, retinas were incubated with the specific primary antibodies diluted in blocking buffer overnight at 4 °C. Retinas were washed three times in PBST and 30 min at room temperature with Pblec (1-mM $MgCl_2$, 1-mM $CaCl_2$, 0.1-mM $MnCl_2$, 1% Triton X-100 in PBS). Retinas were incubated for 2 h at room temperature or overnight at 4 °C with Isolectin B4 (IB4, Invitrogen, #I21412, diluted 1:300) and the corresponding secondary antibodies in Pblec. Subsequently, the retinas were washed three times with PBST and flat-mounted on microscope glass slides with Mowiol/DABCO. For IF stainings of the Golgi (anti-GOLPH4) and EC nuclei (anti-ERG) in retinas (both primary antibodies are raised in the same species), an extra step was performed[82]. After incubation with anti-ERG

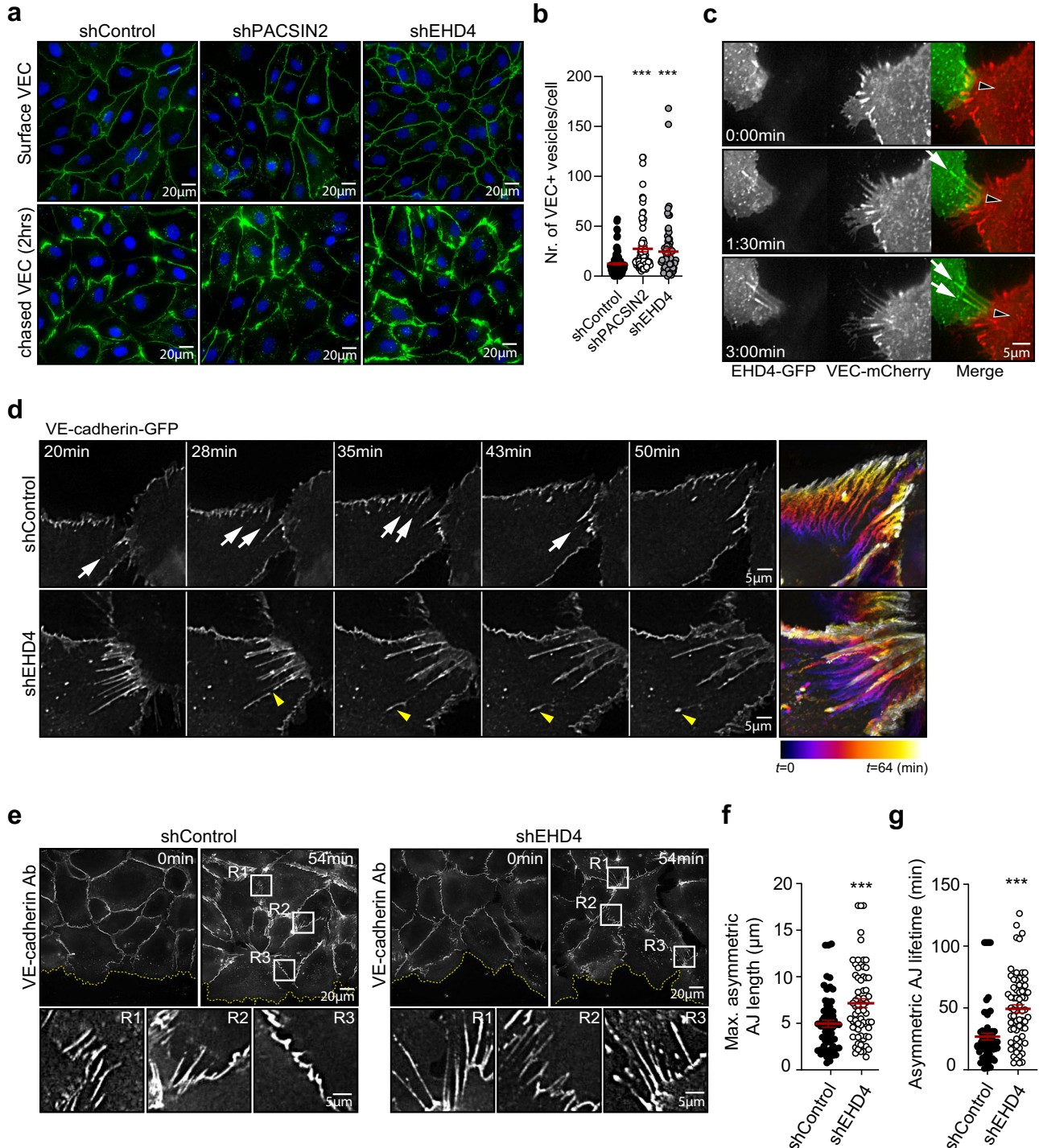

antibody and its corresponding secondary, retinas were blocked for 2 h at RT with the AffiniPure F(ab')2 fragments Donkey anti-rabbit IgG (1/100, Cat# 711-006-152) from Jackson ImmunoResearch. Retinas were washed three times in PBST and fixed with 4% PFA at RT for 5 min. Next, retinas were washed with PBS and blocked with blocking buffer for 30 min at 4 °C followed by incubation with anti-GOLPH4 antibody in blocking buffer overnight at 4 °C. Finally, retinas were incubated with the corresponding secondary antibody in Pblec as described above.

**Flow cytometry.** HUVECs transduced with shControl, shPACSIN2, or shEHD4 were washed with PBS and detached with accutase on ice, and washed with FACS buffer (0.1% BSA, 2-mM EDTA in PBS). Cells were incubated with anti-surface VE-cadherin Alexa Fluor-647 conjugated antibody for 30 min at 4 °C and analyzed on a CytoFLEX S Flow cytometer (Beckman Coulter Life Sciences).

**Fluorescence microscopy.** Standard IF stainings were imaged on a NIKON Eclipse TI widefield microscope that was equipped with a lumencor SOLA SE II light source, standard DAPI, CFP, GFP, mCherry, or Cy5 filter cubes, 10x CFI Achromat dry objective (0.25 NA) or 60x Apo TIRF oil objective (1.49 NA), and an Andor Zyla 4.2 plus sCMOS camera (Figs. 2g, j, 3e, 4h, i, 5, 6a and Supplementary Figs. 4–7). Imaging of retinal stainings was performed on a LEICA SP5 confocal equipped with 40x (1.25 NA) and 63x oil (1.40 NA) objectives and on a Leica TCS SP8 confocal laser scanning microscope, with 10x, 40x oil (1.30 NA) and 63x oil (1.40 NA) objectives using 405-nm UV diode and 470–670-nm White Light lasers (Figs. 1, 2m, 7f, h, j and Supplementary Figs. 1a, 2, 8a). Colocalization studies in Figs. 3b and 4b were performed with the SP8 confocal and 60× objective. Live cell imaging was performed on HUVECs, which were seeded on Lab-Tek chambered 1.0 borosilicate coverglass slides coated with 5-μg ml⁻¹ fibronectin. For live cell imaging, we used the previously specified inverted NIKON Eclipse TI microscope equipped with perfect focus system and Okolab cage incubator and humidified

**Fig. 6 EHD4 depletion perturbs VE-cadherin trafficking and asymmetric AJ remodeling during collective migration. a** Widefield IF images of shControl, shPACSIN2 (mix of D11 + E1), or shEHD4 (mix of E1 + E10) HUVECs that are pulse-labeled with anti-VE-cadherin antibodies (green) and stained for nuclei (blue). Upper row represents the surface-labeled levels of VE-cadherin at $t = 0$ h, and the lower row images represent 2 h chased VE-cadherin. **b** Quantification of the number of VE-cadherin-positive vesicles per cell. The data are from three independent experiments; shControl ($n = 87$ cells), shPACSIN2 ($n = 80$ cells), and shEHD4 ($n = 70$ cells). $P < 0.0001$ when comparing shControl to shPACSIN2 and $P = 0.0001$ when comparing shControl to shEHD4. **c** Widefield time-lapse images of asymmetric AJs in HUVECs expressing EHD4-GFP (green) and VE-cadherin-mCherry (red). White arrows point to EHD4 recruitment, and the black filled arrowheads indicate the displacement direction of the AJs. See Supplementary Movie 3. **d** Widefield time-lapse images of asymmetric AJs in shControl or shEHD4 (mix of E1 + E10) HUVECs expressing VE-cadherin-GFP. White arrows point to gradually turning over of asymmetric AJs through subtle internalization events. Yellow arrowhead highlights the breaking of elongated asymmetric AJ in shEHD4 HUVECs. Heat map in the right image panels shows the corresponding junction dynamics over 64 min in a unique color per time frame. Note the gradual asymmetric AJ turnover in shControl HUVECs. See Supplementary Movie 4. **e** Widefield images of HUVEC monolayers transduced with shControl or shEHD4 (mix of E1 + E10) and live-labeled with anti-VE-cadherin antibody (white) at time $t = 0$ and $t = 54$ min post-scratch wound induction. The yellow punctuated line indicates the border of the scratch wound. The white squares indicate ROIs of remodeling asymmetric AJs between migrating HUVECs. See Supplementary Movie 5 for time-lapse images of the wound closure. **f** Quantification of the maximal length of asymmetric AJs during the time-lapse recordings scratch wound migration (54 min). The data are from three independent experiments; shControl ($n = 63$ asymmetric AJs) and shEHD4 ($n = 68$ asymmetric AJs). $P = 0.0005$ when comparing shControl to shEHD4. **g** Quantification of the lifetime of asymmetric AJs. The lifetime was defined as the time between asymmetric AJ formation and the time of their breakdown. The data are from three independent experiments; shControl ($n = 63$ asymmetric AJs) and shEHD4 ($n = 68$ asymmetric AJs). $P < 0.0001$ when comparing shControl to shEHD4. All graphs represent mean ± SEM (error bars) and the statistical analysis was performed by a Kruskal–Wallis and Dunnett's multiple comparisons test (**b**) or unpaired two-tailed *t*-test (**f, g**). ***$P < 0.001$. Scale bars—20 and 5 μm. VEC VE-cadherin, *t* time, Ab antibody, R region of interest, AJ adherens junction, Nr number. Source data are provided as a Source Data file.

$CO_2$ gas chamber maintaining 37 °C and 5% $CO_2$; (Figs. 2d, g, 6c–e, 7a and Supplementary Movie 1, 3–6); or a Leica TCS SP8 SMD confocal laser scanning microscope (Leica Microsystems), equipped with case incubator maintaining 37 °C and 5% $CO_2$, CS2 63 × /1.40 oil objective and 470–670-nm white light lasers (Supplementary Fig. 3a and Supplementary Movie 2).

**Immunolabeling for live imaging and VE-cadherin pulse-chase experiments.** For live imaging of VE-cadherin experiments, HUVECs were labeled with an Alexa Fluor-647-conjugated Mouse Anti-Human CD144 non-blocking anti-extracellular antibody (Clone 55-7H1; Cat # 561567) diluted 1/200 in EGM-2 at 37 °C. The HUVECs were labeled 10–15 min prior to imaging and maintained in antibody-containing media throughout the imaging experiment (Fig. 6e). For VE-cadherin pulse-chase experiments (Fig. 6a), we pre-labeled the cells with the antibody for 30 min at 4 °C, washed the cells with PBS, and either fixed the HUVECs immediately in 4% PFA in PBS++, or first cultured the cells at 37 °C for 2 h.

**Wound-healing assay.** HUVECs were cultured on 24-well plates or coverslips coated with 5-μg ml⁻¹ fibronectin. After reaching confluency, the monolayers were scratched cross-wise with a p200 pipette tip and were washed with EGM-2. For live imaging, the 24-well plates were mounted on an inverted NIKON Eclipse TI microscope equipped with Okolab cage incubator and humidified $CO_2$ gas chamber maintaining 37 °C and 5% $CO_2$. Phase-contrast live imaging of wound closure was performed with 10x CFI Achromat DL dry objective (0.25 NA) and an Andor Zyla 4.2 plus sCMOS camera for 16–18 h with a time interval of 10 or 15 min. For IF imaging, the coverslips were fixed in 4% PFA in PBS++ 5 h after wounding and subsequently immuno-stained. For the competition scratch assays, HUVECs transduced with shControl-RFP or shPACSIN2 and GFP were seeded as mosaic monolayers in a 24-well plate coated with 5-μg ml⁻¹ fibronectin. Scratch wound migration was imaged overnight on an inverted NIKON Eclipse TI microscope with a 20× objective using GFP and mCherry filter cubes (NIKON). The ImageJ plugin for manual tracking was used for single-cell tracking, and the Chemotaxis tool plugin was used to quantify velocity and directionality.

**Particle image velocimetry (PIV) analysis.** Particle image velocimetry (PIV) was performed using PIVlab software implemented in MATLAB[83]. The velocity fields of the time-lapses of wound-healing assays were determined using an interrogation window set to 50 × 50 pixels with 50% overlap, enable clahe window size of 30 pixels, and applied to 10-h recordings with 15-min time interval. The mean velocity correlation length was determined from exponential fitting of correlation curves between neighboring windows using the *v*-component of the velocity using the MATLAB code as in refs. [37,38].

**Sprouting angiogenesis assay.** For the angiogenic sprouting assay[36], HUVECs were resuspended in EGM-2 medium containing 0.1% methylcellulose (4000 cP, Sigma, #M0512). For spheroid formation, 750 cells per 100-μl methylcellulose medium were seeded in wells of a U-bottom 96-wells suspension plate and incubated overnight. Next, glass-bottom 96-well plates were coated with 50 μl/well of 1.7-mg ml⁻¹ Type I rat tail collagen (IBIDI, #50201) mixed with FCS and EGM-2 and placed at 37 °C for 30 min. Then, spheroids were collected and resuspended in the collagen mixture and plated 50 μl/well on top of the coated glass-bottom 96-well plates and placed at 37 °C. After polymerization of the collagen gel, spheroids

were stimulated with 50-ng ml⁻¹ VEGF to induce sprouting overnight. Pictures were acquired using an EVOS M7000 imaging system and 10× objective. Sprouting number and length was analyzed using ImageJ.

**Co-IP assay and Western blot analysis.** Co-IP on untransduced HUVECs, HUVECs ectopically expressing GFP, GFP-tagged PACSIN2 or EHD4 was performed with magnetic GFP-Trap agarose beads from Chromotek (Cat # gtma-20). Prior to the procedure, the magnetic beads were washed three times in PBS, two times in lysis buffer, blocked for 1 h with 2% BSA in PBS at 4 °C and washed one time with lysis buffer. Cells were lysed with freshly made ice-cold lysis buffer (20-mM HEPES pH 7.4, 150-mM NaCl, 1,7-mM $CaCl_2$, and 0.5% NP-40 supplemented with cOmplete™, Mini, EDTA-free Protease Inhibitor Cocktail (Sigma-Aldrich)). The lysates were homogenized with a Douncer homogenizer and sonicated thoroughly. The crude cell lysates were then incubated on ice for 30 min and centrifuged at 12,000 $g$ and 4 °C for 15 min. A portion of the supernatant was used as whole lysate for Western Blot analysis. The rest of the supernatant was added to the GFP-Trap-coupled beads, incubated overnight at 4 °C and then washed three times with lysis buffer and PBS on ice. Immunoprecipitates were dissolved in reduced sample buffer, boiled for 5 min and analyzed by Western Blot. Western Blot analysis was performed according to standard Western Blot technique protocols. All samples were taken up in reduced sample buffer. Images were acquired using ImageQuant LAS4000 mini. Source data are provided as a Source Data file.

**Image analysis and quantification.** Images were acquired using Nikon Imaging Software Elements and Leica Application Suite. Images were enhanced for display with an unsharp mask filter or processed in ImageJ/Adobe Photoshop. All image analyses for quantifications were performed in ImageJ. Overview and high-resolution images of retinal IF are maximum intensity projections. Quantifications were performed on the obtained high-resolution confocal images as follows: endothelial branch points and proliferating cells were quantified behind the angiogenic front in image field sizes of 100 × 100 μm. The number of sprouts, length of the sprouts, number of endothelial cells, and clustering of cells were quantified at the angiogenic front. The percentage of proliferating cells per unit area in the sprouting front was based on cytoplasmic (dividing cells) versus nuclear ERG (non-dividing cells) stainings. The total number of sprouts and the number of endothelial cells was determined per 100 μm of the angiogenic front border. The number of sprouting clusters was defined as three or more cells per sprout. The number of ESM1⁺ cells at the sprouting front was calculated by dividing the number of ESM1⁺ endothelial cells per number of total endothelial cells at the sprouting front. VE-cadherin cytoplasmic intensity at the sprouting front was determined by measuring the mean intensity of multiple fixed areas of interest inside the vessels per image and subtracting the mean of the background signal outside the vessels. Golgi orientation in retinas was measured by relating the position of GOLPH4 signal to the center of the mass of the nucleus (ERG) in relation with the vascular sprouting front and represented in a Rose plot showing the angular distribution of individual Golgi orientation (performed in R). The percentage of endothelial cells in the first three rows (tip and stalks cells) with the Golgi polarized ±60° toward the sprouting front direction was determined. In images from scratch wound assays, the boundaries of the wound were manually marked and the decrease of the wounded surface area was measured over time to quantify wound closure velocity. Golgi orientation was assessed by relating the positioning of the GM130 signal to the center of mass of nucleus (DAPI) in relation

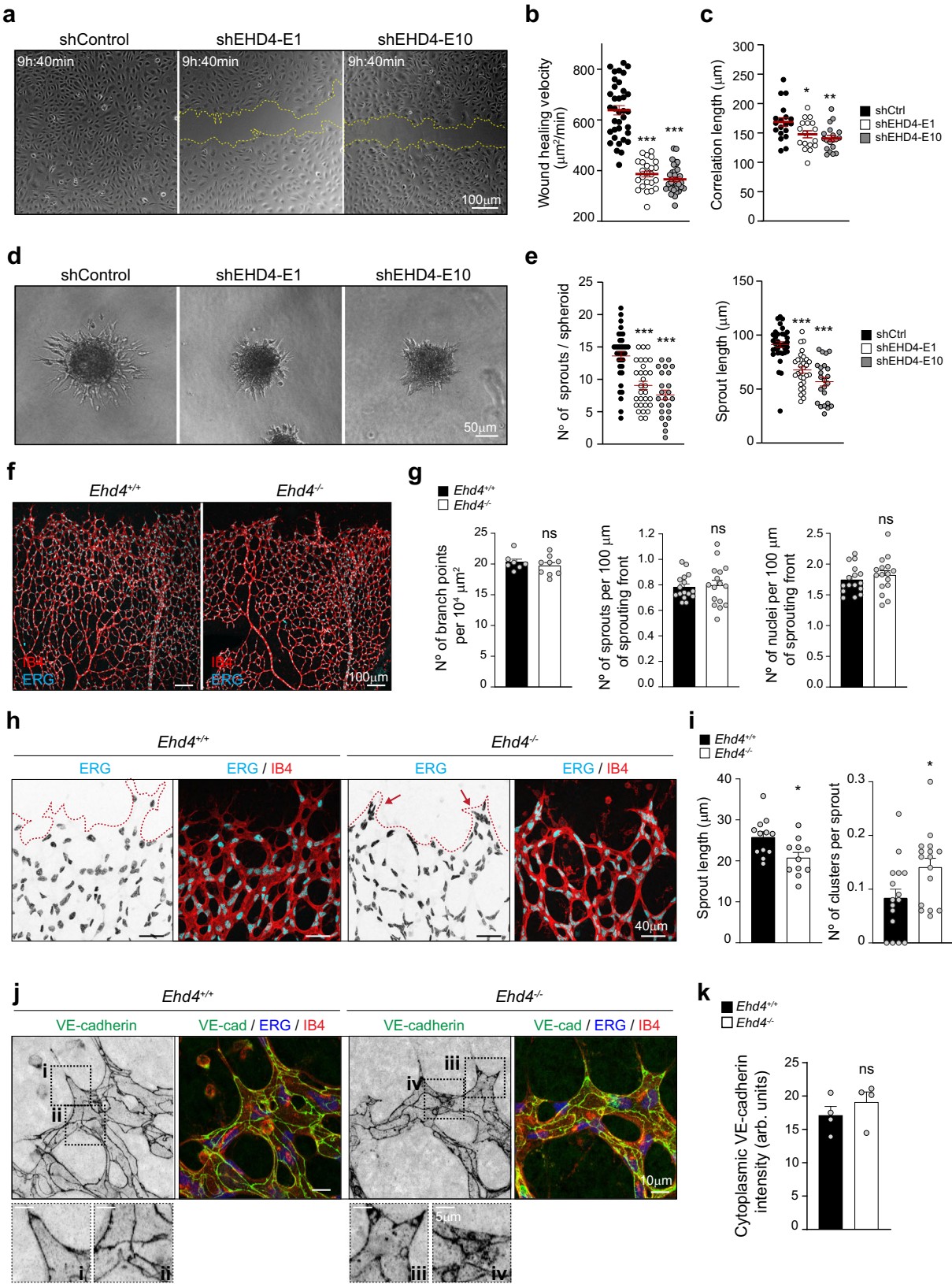

to the scratch direction. Linescan analysis was performed using 3-μm long lines that span asymmetric AJs measured the integrated intensity profile of the imaged proteins. These profiles were background corrected with the integrated intensities in 3-μm linescans in the cytoplasm in close proximity to the junction. Quantifications of the percentage of PACSIN2, EHD4, or MICAL-L1 positive junctions

were calculated as the ratio between all manually counted asymmetric AJs and asymmetric AJs featuring the presence of tubular PACSIN2, EHD4, or MICAL-L1 signals, respectively. Colocalization of proteins at asymmetric AJs were assessed at regions of interest around the tubular junctional structures and defined by the Pearson's colocalization coefficient obtained by the ImageJ colocalization plugin

**Fig. 7 EHD4 is needed for endothelial directed migration, in vitro angiogenic sprouting and coordination of angiogenesis in vivo. a** Representative phase-contrast images of scratch wound assays at 9 h and 40 min post-scratch performed on HUVEC monolayers transduced with shControl, shEHD4-E1, and shEHD4-E10. The punctuated yellow lines indicate the boundaries of the wound. See Supplementary Movie 6 for time-lapse images of the scratch wound migration. **b** Quantification of wound-healing velocity measured in surface area per min of post-scratch HUVEC monolayers transduced with shControl ($n = 36$ movies), shEHD4-E1 ($n = 24$ movies) and shEHD4-E10 ($n = 35$ movies) from three independent experiments. $P < 0.0001$ when comparing shControl to shEHD4-E1 and when comparing shControl to shEHD4-E10. **c** Quantification of the correlation length of wound-healing time-lapse recordings of HUVECs transduced with shControl ($n = 19$ movies), shEHD4-E1 ($n = 17$ movies), and shEHD4-E10 ($n = 19$ movies) from four independent experiments using particle image velocimetry (PIV) analysis. $P = 0.0384$ when comparing shControl to shEHD4-E1 and $P = 0.0035$ when comparing shControl to shEHD4-E10. **d** Representative phase-contrast images of sprouting spheroids from HUVECs transduced with shControl, shEHD4-E1, and shEHD4-E10 after 16-h stimulation with VEGF. **e** Quantification of the number of sprouts per spheroid and the average sprout length of HUVECs transduced with shControl ($n = 35$ spheroids), shEHD4-E1 ($n = 30$ spheroids), and shEHD4-E10 ($n = 23$ spheroids). Data are from three independent experiments. $P < 0.0001$ when comparing shControl to shEHD4-E1 and when comparing shControl to shEHD4-E10. **f** Representative images of whole-mount retinas stained for Isolectin B4 (IB4, red) and ERG (cyan) from control ($Ehd4^{+/+}$) and $Ehd4^{-/-}$ mouse littermates at P6. **g** Quantification of the number of branch points per unit area (at least seven retinas per genotype from three independent littermates), the number of sprouts and the number of nuclei per 100 μm of sprouting front border (at least 16 retinas per genotype from at least three independent littermates) in the sprouting front of control and $Ehd4^{-/-}$ P6 retinas. **h** Representative images of magnified sprouting front of retinas from control and $Ehd4^{-/-}$ mice at P6 stained for ERG (cyan) and IB4 (red). Red punctuated lines indicate the sprouting front boundary. The red arrows indicate nuclear clusters in abnormal sprouts. **i** Quantification of the average sprout length and the number of endothelial cell clusters at the sprouting front of control and $Ehd4^{-/-}$ P6 retinas (at least 11 retinas per genotype from at least three independent littermates). $P = 0.0268$ when comparing the sprout length and $P = 0.0290$ when comparing the number of endothelial clusters in $Ehd4^{+/+}$ to $Ehd4^{-/-}$ retinas. **j** Representative high-resolution images of the sprouting front from control and $Ehd4^{-/-}$ retinas stained for VE-cadherin (green), ERG (blue), and IB4 (red). The black dotted squares indicate the ROIs that are magnified in the panels below. **k** Quantification of the intensity of cytoplasmic VE-cadherin at the vascular sprouts of control and $Ehd4^{-/-}$ retinas ($n = 4$ retinas per genotype, from two independent littermates). All quantifications represent mean ± SEM (error bars), and statistical analysis was performed by a one-way ANOVA and Dunnett's multiple comparisons test in (**b**, **c**, **e**) and two-sided Mann–Whitney test in (**g**, **i**, **k**). ns non-significant; *$P < 0.05$; **$P < 0.01$; ***$P < 0.001$. Scale bars 100 μm (**a**, **f**), 50 μm (**d**), 40 μm (**h**), 10 and 5 μm (**j**). Arb. units arbitrary units. Source data are provided as a Source Data file.

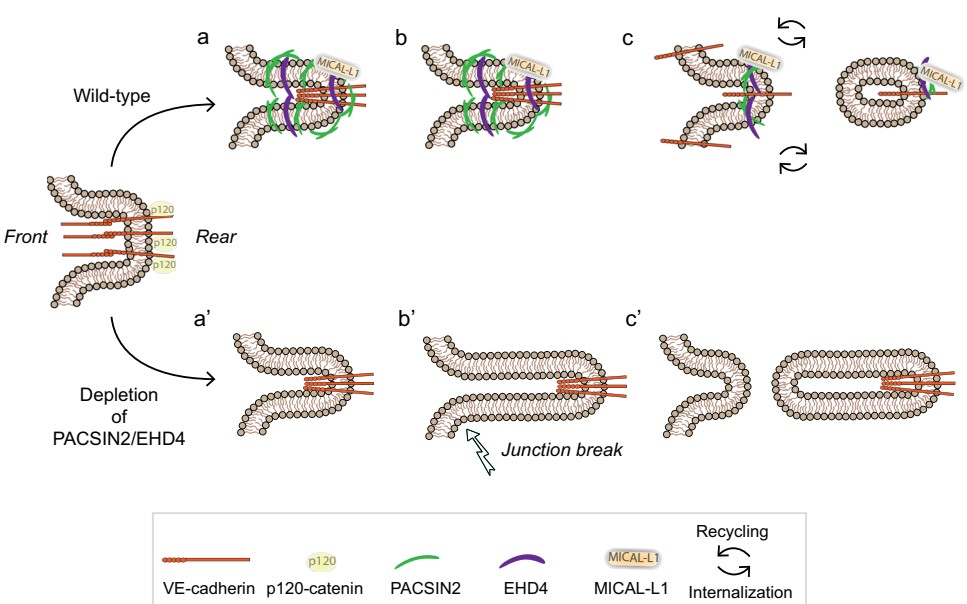

**Fig. 8 Schematic representation of how the PACSIN2/EHD4/MICAL-L1 complex controls trafficking of VE-cadherin at asymmetric junctions.** Asymmetric adherens junctions are formed between leader and follower cells in migrating endothelial collectives. The F-BAR protein PACSIN2 as well as the trafficking and recycling proteins EHD4 and MICAL-L1 are recruited to the rear of asymmetric junctions (**a**). The PACSIN2/EHD4/MICAL-L1 complex enables the proper trafficking of VE-cadherin at the junction rear, ensuring gradual turnover of asymmetric junctions in support of collective migration (**b**, **c**). In the absence of PACSIN2 or EHD4, this VE-cadherin trafficking systems fails, leading to asymmetric junction elongation and eventually ruptures (**a'**, **b'**), resulting in an accumulation of internalized VE-cadherin levels (**c'**).

JACOP. The number of VE-cadherin-positive vesicles in pulse-chase experiments was quantified as the number of vesicles detected in the cytoplasm using the Find Maxima (prominence = 15000) and Analyze Particle (size = 0.1–1.5; circularity = 0.50–1.00) tools in ImageJ. Asymmetric AJs maximal length and lifetime during wound closure were quantified by manually tracking remodeling AJs in the first two rows of migrating cells from the moment of their formation. The intensity of Western Blot bands was measured by the Gel Analyzer plugin in ImageJ.

**Statistics**. All data were analyzed using GraphPad Prism software. The graph error bars represent mean ± SEM. Sample size and experimental replicates are indicated in the figure legends. We used the nonparametric two-sided Mann−Whitney's test, the parametric Student's $t$ test, one-way ANOVA or Kruskal–Wallis with Dunnett's multiple comparison test, and a paired nonparametric Wilcoxon test. We designated $P$ values in figures as: ns not significant; *$P < 0.05$; **$P < 0.01$; ***$P < 0.001$.

**Reporting summary**. Further information on research design is available in the Nature Research Reporting Summary linked to this article.

## Data availability

Source image data underlying Figs. 2a, 4d, f, 5b, and Supplementary Figs. 1c, d, e, 3b, 8c are provided as a Source Data File. The numerical data underlying Figs. 1c, d, f, 2c, e, f, h, i, l, n, o, 3c, d, g, 4c, e, g, j, k, l, 5b, d, f, 6b, f, g, 7b, c, e, g, i, k and Supplementary Figs. 4–6 are provided as a Source Data File. Additional datasets generated in the current study are available from the corresponding author on request. Transgenic mice or detailed information to generate them are available via M.P. Plasmids and other biological materials are available from the corresponding author. Source data are provided with this paper.

## Code availability

Data analysis was performed with published and publicly available software.

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

## Acknowledgements

We are grateful to Prof. Dr Steve Caplan (University of Nebraska Medical Center, US) for providing the peGFP-EHD4 plasmid, Prof. Dr René-Marc Mège for sharing expertise with the MATLAB correlation code (Paris Diderot University, France), and Dr Peter Stroeken (Amsterdam UMC, University of Amsterdam, the Netherlands) for providing lentiviral shRNAs from the MISSION library (Sigma-Aldrich) of the RNAi Consortium. We thank the Cellular Imaging core facility of the Amsterdam UMC for technical support. The group of S.H. is financially supported by the Netherlands Organization of Scientific Research (ZonMW VIDI Grant 016.156.327 and NWO OCENW.KLEIN.281). M.P. is supported by the German Research Foundation (PL 233/3-3). IDIBELL is a member of Centres de Recerca de Catalunya (CERCA), and M.G. is supported by research Grants SAF2017-89116R-P from MCIU (Spain) co-funded by European Regional Developmental Fund (ERDF) a Way to Build Europe, La Caixa Foundation (HR18-00120), la Asociación Española contra el Cancer (AECC)-Grupos Traslacionales (GCTRA18006CARR), and la Fundación BBVA.

## Author contributions

T.S.M., A.A.-U., and S.H. conceived and designed the study. T.S.M., A.A.-U., M.M.v.d.S, V.J., A.d.H., A.G.G., M.T., and S.H. designed, performed, collected data, and analyzed the cell-based in vitro experiments. T.S.M., A.A.-U., J.N., M.T., M.G., M.P., and S.H. designed, performed, collected data, and analyzed the mouse-based experiments. T.S.M., A.A.-U., and S.H. wrote the manuscript. All authors discussed the results and reviewed the manuscript.

## Competing interests

The authors declare no competing interests.
