## [Peer Review File · Nature Communications]

REVIEWER COMMENTS

Reviewer #1 (Remarks to the Author):

The manuscript by Malinova and Angulo-Urarte et al reports on the function of a pascin2 complex in endothelial cell migration, polarity and angiogenesis. The authors conclude that PACSIN2 demarcates asymmetric adherens junctions between leader and follower cells during collective migration. In addition, they localize the endocytic recycling regulators EHD4 and MICAL-L1 to the rear end of asymmetric adherens junctions. Finally, the authors indicate that the PACSIN2/EHD4/MICAL-L1 complex regulates the trafficking of VE-cadherin, which in turn regulates polarized endothelial migration and angiogenesis. The data presentation is beautiful, and the authors have explored pascin2/EHD4 functions in cells and in mouse models, and they have utilized a combination of imaging and biochemistry. This is a very interesting manuscript and will have broad appeal. Several weaknesses in interpretation and discussion are outlined below.

1) In Fig. 1 the authors do a nice job quantitatively describing alterations in retinal vasculature in the Pascin2^{-/-} mice. However, the data in Fig. 1D on cytoplasmic accumulation of VE-cadherin is less than convincing. In fact, later in the paper (Discussion) the authors indicate that “this complex guides local trafficking of internalized VE-cadherin towards Rab4 positive endosomes, and keeps internalized VE-cadherin away from lysosomal degradation.” Wouldn't we expect less cytoplasmic VE-cadherin if the cadherin was shunted to a degradative compartment? It would be helpful if the authors assessed relative cell surface and cytoplasmic levels of VE-cadherin in Pascin2 knock down (and/or EHD4 and MICAL-L1 knock down cells) in vitro to understand how this complex is regulating cell surface vs cytoplasmic accumulation of the cadherin.

2) In figure 2 the authors nicely show that loss of pascin2 compromises the ability of endothelial cells to polarize during cell migration. Given the role for pascin2 and related complexes (see below) in endocytic trafficking, the authors put their work in context with a recent study by Grimsley-Myers et al showing that VE-cad endocytosis regulates endothelial polarity during migration and defects in postnatal retinal angiogenesis (PMID: 32232465).

3) In figure 3 the authors conclude that “This indicates that junctional recruitment of PACSIN2 occurs separately from the mechanism driving p120-catenin controlled constitutive VE-cadherin endocytosis.” The data nicely show that VE-cad that is deficient in constitutive endocytosis (VEcadDEE) and/or unable to bind p120 (VEcadGGG) still recruit pascin2. I'm less clear on whether or not loss of pascin2 inhibits constitutive endocytosis or that the VEcadDEE mutant is (or is not) internalizing from the pascin2-positive tubules. In other words, the data don't seem to rule out that pascin2 is involved in constitutive endocytosis or that constitutive endocytosis is involved in the formation of the pascin2 tubular structures.

4) In figure 6, the interpretations of these data are confusing. First, the colocalization of Rab4 and LAMP2 are not convincing. Are these structures (Rab4 and LAMP2 vesicles) associating with the VE-cad tubules? The Rab4.GFP is present all over the cytoplasm, and localization at junctions seems difficult to assess. It's not clear to me that these data add to the story. The authors are concluding

that VE-cad is trafficking normally to a recycling compartment but that in the absence of the pascin/EHD4/MICAL complex, the VE-cad is entering a degradative compartment. The data do not convince this reviewer that a change in destination or turnover of the cadherin is occurring. In fact, the authors state in the intro to figure 6 that “VE-cadherin is rapidly internalized from asymmetric AJs that are devoid of PACSIN2, and subsequently enters Ras-related protein 4 (Rab4) positive recycling endosomes (19).” This statement implies that VE-cad enters a recycling compartment when it is NOT associated with pascin2. It is unclear where the authors think the pascin2 complex is functioning – at the tubular VE-cad structures or separately in a recycling endosome compartment.

5) Along with the comments above, a model diagram describing how the pascin2 complex is regulating VE-cad trafficking would be helpful.

Minor comments:

- a) Secondary antibodies and catalog numbers are not mentioned in methods.
- b) Company/ catalog no. of Mowiol4-88/DABCO is not mentioned.
- c) In the Fig. 1d legend, colors for ERG and IB4 are reversed, although correct in the figure itself.
- d) Fig. 4h, slight formatting error with black outline for F-actin.
- e) Fig. 7f, ERG is blue not green (in the legend).
- f) On page 7, switch Suppl. Figures 5 and 6 (either the numbers or the figures itself) to maintain continuity.

Reviewer #2 (Remarks to the Author):

Collective cell migration is crucial for sprouting angiogenesis. During this process, coordinated rearrangement of ECs-ECs junctions mediated by VE-cadherin is important for the transmission of forces between leading cells and following cells. The authors have previously shown that PACSIN2, the F-BAR protein, is the key regulator of VE-cadherin at the tensile adherens junctions of the trailing ends of the migrating ECs in the monolayer (Dorland et al Nat. Commun 2016). Here in this manuscript, the authors generated PACSIN2 KO mice and analyzed the effect of loss of PACIN2 during sprouting angiogenesis. The authors showed that PACSIN2 coordinates endothelial front-rear polarity in vivo and in vitro. Furthermore, the authors identified that EHD4 and MICAL-L1 forms the protein complex with PACSIN2 to modulate VE-cadherin trafficking in vitro. Finally, the authors confirmed that EHD4 KO mice shows a similar phenotype to PACSIN2 KO mice. The authors are advancing the concept that the molecular mechanisms underlying the transmission of tension at the rear of leading cells is important for the establishment of a front of migrating cells in vivo. Most of experiments were carefully designed and well done. The manuscript is written well and easy to follow. However, I feel the current experimental setting is not sufficient to draw the authors' conclusion. To improve this, the following points should be addressed

Specific comments

1. The analysis and interpretation of the phenotype of PACSIN2 KO mice at angiogenic front is unclear and insufficient. Are the shorter sprouting and clusters of ECs due to the defected polarized cell migration? Is the differentiation or gene transcription of ECs affected? For instance, is the

expression of the tip cell marker OK?

2. Mechanical forces are important for asymmetric behavior of ECs. In vivo, endothelial cells are subjected to shear stress, which triggers directional endothelial migration (Franco et al., Plos Biol 2015). Additionally, ECs migrate not only toward the avascular region, even at the angiogenic front (Pitulescue et al Nat Cell Biol 2017). In figure 2, the author examined the effect of KD on Golgi orientation during the wound healing process. The Golgi orientation in ECs is very good indicator of polarized migration. Is the Golgi orientation affected in the PACSIN2 and EHD4 KO mice in regard to the polarized migration?

3. Is the barrier function of EC junction affected by PACSIN2 KO mice?

4. Collective migration is a team effort for many cells. The question is whether the effect of PACSIN2 KD is cell-autonomous or not. Mosaic experiment in vitro would be interesting.

5. Where is PACSIN2 expressed in vivo? Are they specifically expressed at EC junction sites? Given the fact that the authors used global KO mice, this information is necessary. Additionally, if they use EC specific PACSIN2 KO mice, do they observe the similar phenotype with global KO mice?

6. In their previous manuscript, the authors revealed the role of PACSIN2 on VE-cadherin tensile junction, which is asymmetrically existing migrating ECs. While VE-cadherin is localized at all endothelial cell-to-cell contact sites, does PACSIN2 colocalize with VE-cadherin asymmetrically in vivo?

7. It seems that PACSIN3 expression is increased in the retina of PACSIN2 KO mice. Does this cause functional redundancy and rather weak phenotype of PACSIN2 KO mice? Is PACSIN3 expression increased in ECs?

8. In figure 1d, staining intensity against VE-cadherin seems to be increased not only in the cytoplasmic region but at cell-cell contact sites. This is not mentioned.

9. Is the level of cytoplasmic VE-cadherin increased in EHD4 KO mice?

10. Is the increased level of cytoplasmic VE-cadherin observed in vitro as well?

11. In Supplementary Fig1d, Ponceau S staining suggests the protein loading for western blotting analysis is not same.

Response to the Reviewers

Reviewer #1 (Remarks to the Author):

The manuscript by Malinova and Angulo-Urarte et al reports on the function of a pascin2 complex in endothelial cell migration, polarity and angiogenesis. The authors conclude that PACSIN2 demarcates asymmetric adherens junctions between leader and follower cells during collective migration. In addition, they localize the endocytic recycling regulators EHD4 and MICAL-L1 to the rear end of asymmetric adherens junctions. Finally, the authors indicate that the PACSIN2/EHD4/MICAL-L1 complex regulates the trafficking of VE-cadherin, which in turn regulates polarized endothelial migration and angiogenesis. The data presentation is beautiful, and the authors have explored pascin2/EHD4 functions in cells and in mouse models, and they have utilized a combination of imaging and biochemistry. This is a very interesting manuscript and will have broad appeal. Several weaknesses in interpretation and discussion are outlined below.

We kindly thank the Reviewer for the positive and insightful evaluation. We have addressed the remaining points as follows:

1) In Fig. 1 the authors do a nice job quantitatively describing alterations in retinal vasculature in the *Pascin2*^{-/-} mice. However, the data in Fig. 1D on cytoplasmic accumulation of VE-cadherin is less than convincing. In fact, later in the paper (Discussion) the authors indicate that “this complex guides local trafficking of internalized VE-cadherin towards Rab4 positive endosomes, and keeps internalized VE-cadherin away from lysosomal degradation.” Wouldn’t we expect less cytoplasmic VE-cadherin if the cadherin was shunted to a degradative compartment? It would be helpful if the authors assessed relative cell surface and cytoplasmic levels of VE-cadherin in *Pascin2* knock down (and/or EHD4 and MICAL-L1 knock down cells) *in vitro* to understand how this complex is regulating cell surface vs cytoplasmic accumulation of the cadherin.

We agree with the Reviewer that the trafficking of internalized VE-cadherin molecules in the various model systems was not fully clarified yet. In the experiments displayed in Fig. 1 we first investigated the physiological importance of PACSIN2 for retinal angiogenesis, a crucial developmental process that depends on turnover of VE-cadherin. In addition, we have imaged VE-cadherin in the developing vasculature to assess whether the angiogenic defects in PACSIN2 knockouts is correlated with changes in VE-cadherin organization. Other well-respected angiogenesis labs, who study VE-cadherin dynamics in the mouse retinal angiogenesis model at this high resolution (Bentley et al., 2014; Carvalho et al., 2019), also determined the cytoplasmic pool of VE-cadherin as an important feature of endothelial remodeling sections of the angiogenic front. Doing so, we consistently observed that the intensity of cytoplasmic VE-cadherin is higher in the PACSIN2 knockout retinas.

We have previously shown that total VE-cadherin cell surface levels do not change upon PACSIN2 knockdowns, but that the depletion of PACSIN2 does promote the level of internalized VE-cadherin in vesicles ((Dorland et al., 2016) Fig. 9d of that paper), which likely encompass both recycling, early and degradative endosomes. This indicates that PACSIN2 controls the trafficking of VE-cadherin from the cell’s surface to cytosolic vesicles. By employing FRAP experiments and VE-cadherin internalization assays, we further demonstrated that PACSIN2 functions locally to control the turnover of the VE-cadherin complex at the trailing ends of the asymmetric junctions (Dorland et al., 2016). Thus, our current finding that cytoplasmic VE-cadherin signal is higher in the angiogenic endothelium of PACSIN2 knockout retinas is in strong agreement with the effect that PACSIN2 depletion has on *in vitro* cultured endothelial cells. We speculate that the difference in the window of PACSIN2-controlled cytoplasmic VE-cadherin levels *in vivo* (in which we see a slight effect) vs. *in vitro*, relate to the notion that the PACSIN3 isoform compensates for the loss of PACSIN2 in the *in vivo* knockout model (see also our response to Reviewer 2).

Based on the Reviewers' suggestion, and to gain more insight into how the PACSIN2/EHD4 complex controls VE-cadherin trafficking, we performed new experiments to assess the surface levels of VE-cadherin in shControl, shPACSIN2 and shEHD4 HUVECs by FACS analysis. These new experiments demonstrate that total surface levels of VE-cadherin are unaltered by their knockdowns (Suppl Fig. 7b). Next, we determined VE-cadherin intensity and subcellular localization in shControl, shPACSIN2, and shEHD4 monolayers by selectively visualizing the VE-cadherin molecules on the cell surface by immunostainings without permeabilization and using a non-blocking anti-VE-cadherin antibody that recognizes an epitope on the extracellular domain of VE-cadherin. In agreement with the FACS data and our earlier publication (for shPACSIN2 HUVECs; (Dorland et al., 2016)), we observed no differences in VE-cadherin localization or intensity in the various knockdown conditions compared to control (Fig. 6a upper images), indicating that global VE-cadherin-based junction biogenesis pathways are unaffected. Hence, the action of the PACSIN2/EHD4 complex in VE-cadherin trafficking likely takes place locally at the fraction of asymmetric junctions to which the PACSIN2 and EHD4 proteins are recruited to.

To study if EHD4, like PACSIN2 (Dorland et al., 2016), controls VE-cadherin trafficking we pulse-labelled VE-cadherin molecules on the surface of shControl, shPACSIN2 and shEHD4 HUVECs and followed their turnover and internalization. These experiments show that in a time course of 2 hrs a larger proportion of VE-cadherin is internalized and accumulating in intracellular vesicles upon depletion of PACSIN2 or EHD4 (New Fig. 6a, b). Taken together, these new results demonstrate that the PACSIN2/EHD4 complex enables coordinated VE-cadherin trafficking during the remodeling of endothelial cell-cell junctions.

2) In figure 2 the authors nicely show that loss of pascin2 compromises the ability of endothelial cells to polarize during cell migration. Given the role for pascin2 and related complexes (see below) in endocytic trafficking, the authors put their work in context with a recent study by Grimsley-Myers et al showing that VE-cad endocytosis regulates endothelial polarity during migration and defects in postnatal retinal angiogenesis (PMID: 32232465).

Indeed we already anticipated that comparable molecular mechanisms might be controlled by PACSIN2, hence we tested this hypothesis directly by investigating junctional PACSIN2 recruitment in endothelial cells expressing the VE-cadherin DEE and GGG mutated versions of VE-cadherin that prevent or enhance VE-cadherin endocytosis respectively, which we based on the original (Nanes et al., 2017) paper from the Kowalczyk lab. Our experiments with these mutants show that PACSIN2 recruitment to the endothelial junctions can occur even in the absence of p120-catenin binding to VE-cadherin, and in the absence of a functional endocytic motif in the VE-cadherin cytoplasmic tail. This prompted us to further investigate how PACSIN2 might accomplish its effects on VE-cadherin-based junction turnover, which led to the discovery of EHD4 and MICAL-L1 as novel protein trafficking controllers of asymmetric junctions during collective cell migration and angiogenesis.

The recent Grimsley-Myers 2020 study (from the same Kowalczyk lab), which was published while our manuscript was under review at Nature Communications, elegantly established that the p120-mediated turnover of VE-cadherin controls endothelial collective migration and endothelial cell polarization. Our finding that the PACSIN2-EHD4-mediated endothelial junction turnover drives endothelial collective migration and angiogenesis is in strong agreement with the notion that proper trafficking of VE-cadherin is needed for endothelial polarity and retinal angiogenesis. Given these timely new exiting findings and complementary insights from both studies, we included the following sentences to the discussion section: "Recent experiments using transgenic VE-cadherin knock-in mouse models, established that the p120-mediated turnover of VE-cadherin is important for endothelial collective migration and endothelial cell polarization (Grimsley Meyer et al.). Our current finding that the PACSIN2-EHD4 complex controls VE-cadherin turnover and drives endothelial collective migration and angiogenesis is in strong agreement with the notion that proper trafficking of VE-cadherin is needed for endothelial polarity and retinal angiogenesis."

3) In figure 3 the authors conclude that “This indicates that junctional recruitment of PACSIN2 occurs separately from the mechanism driving p120-catenin controlled constitutive VE-cadherin endocytosis.” The data nicely show that VE-cad that is deficient in constitutive endocytosis (VEcadDEE) and/or unable to bind p120 (VEcadGGG) still recruit pascin2. I’m less clear on whether or not loss of pascin2 inhibits constitutive endocytosis or that the VEcadDEE mutant is (or is not) internalizing from the pascin2-positive tubules. In other words, the data don’t seem to rule out that pascin2 is involved in constitutive endocytosis or that constitutive endocytosis is involved in the formation of the pascin2 tubular structures.

This is correct, and we like to thank the Reviewer for pointing out this issue. Based on our experiments we can conclude that the recruitment of PACSIN2 technically can occur separately from the release of p120-catenin from the cytoplasmic tail of VE-cadherin if cells are genetically modified, but we cannot rule out that there is no functional association with the downstream constitutive VE-cadherin endocytosis pathway. Furthermore, we show that in the wild-type situation p120-catenin first dissociates from VE-cadherin, which is subsequently followed by PACSIN2 recruitment (Fig 3b-d, Suppl Movie 2). This suggests that in normal endothelial cells the release of p120-catenin from VE-cadherin triggers endocytosis at asymmetric junctions prior to PACSIN2 recruitment. In the revised manuscript we have changed our statements regarding the implications of the results with the VE-cadherin DEE and GGG mutants (Fig. 3e-g) and conclude that: “PACSIN2 recruitment occurs in parallel with the mechanism driving p120-catenin controlled VE-cadherin endocytosis”. This conclusion is based on the finding that junctions that are formed by the p120-binding deficient VE-cadherin mutants still recruit PACSIN2 [Fig. 3e], and the notion that PACSIN2 is recruited to junctions via the interaction of its F-BAR domain with the junctional membrane (Dorland et al., 2016), rather than binding to the VE-cadherin cytoplasmic domain like p120-catenin itself does. In the discussion of the revised manuscript we now further mention that “Our data reveal that PACSIN2 is recruited to asymmetric AJs that are primed for endocytosis by the dissociation p120-catenin from the VE-cadherin complex.”, to emphasize the connection between these molecular events at VE-cadherin-based junctions.

4) In figure 6, the interpretations of these data are confusing. First, the colocalization of Rab4 and LAMP2 are not convincing. Are these structures (Rab4 and LAMP2 vesicles) associating with the VE-cad tubules? The Rab4.GFP is present all over the cytoplasm, and localization at junctions seems difficult to assess. It’s not clear to me that these data add to the story. The authors are concluding that VE-cad is trafficking normally to a recycling compartment but that in the absence of the pascin/EHD4/MICAL complex, the VE-cad is entering a degradative compartment. The data do not convince this reviewer that a change in destination or turnover of the cadherin is occurring. In fact, the authors state in the intro to figure 6 that “VE-cadherin is rapidly internalized from asymmetric AJs that are devoid of PACSIN2, and subsequently enters Ras-related protein 4 (Rab4) positive recycling endosomes (19).” This statement implies that VE-cad enters a recycling compartment when it is NOT associated with pascin2. It is unclear where the authors think the pascin2 complex is functioning – at the tubular VE-cad structures or separately in a recycling endosome compartment.

This issue raised by the Reviewer partially overlaps with the first comment. We sincerely apologize for not having been sufficiently clear in our initial explanation. Of note, part of the trafficking questions in regard to the function of PACSIN2 were already addressed in our previous study where we showed that the fraction of VE-cadherin that is endocytosed from asymmetric AJs rapidly enters Rab4- and Rab5 positive endosomes that are located in close vicinity of the respective junctions. Upon depletion of PACSIN2 with shRNAs the amount of surface-chased VE-cadherin accumulating in cytoplasmic vesicles is strongly enhanced. Which likely includes both early endosomes (Rab5 or EEA1 positive), recycling endosomes (Rab4 positive) and other trafficking compartments. Because PACSIN2 is precisely demarcating the boundaries of the AJ structure and the internalizing VE-cadherin pool we suspected that the newly identified PACSIN2/EHD4/MICAL-L1 complex functions locally at the tubular-shaped asymmetric junctions. This concept is corroborated by the finding that the asymmetric junctions in scratch wound migration are not properly turning over upon depletion of EHD4 (Fig. 6 e-g).

In the revised manuscript, we have changed the text of the introduction of Fig. 6 to better explain this, in line with our previous data and current new findings: “Asymmetric AJs are remodelled in a front-rear polarized fashion, and local PACSIN2 recruitment protects the integrity of VE-cadherin-based junctions (Cao et al., 2017; Dorland et al., 2016). Concordantly, the depletion of PACSIN2 leads to strongly augmented VE-cadherin internalization levels (Dorland et al., 2016).”

We understand that the Reviewer was not convinced about the junctional localization of Rab4-GFP and LAMP2 in shControl vs. shEHD4 HUVECs due to the abundant presence of Rab4 and LAMP2 in the cytoplasm. Of note, in the original submitted manuscript, we analyzed Pearson’s correlation at the asymmetric junction regions only, which indicated that in the absence of EHD4, Rab4-GFP signal does not colocalize with VE-cadherin at asymmetric junctions anymore. To try to gain more insights in the possible association of recycling compartments directly at the asymmetric junctions, we have performed new VE-cadherin pulse-chase experiments in conjunction with immunostainings for endogenous Rab35, the only recycling Rab GTPases for which good working antibodies exist. Although we did find association of Rab35 with the trailing end of asymmetric junctions (Fig. i for Reviewers), the experiments did not show a consistent interaction with asymmetric junctions in a similar fashion as PACSIN2, EHD4 and MICAL-L1 do.

Figure i : Image of asymmetric adherens junctions in HUVECs immunostained for VE-cadherin and the recycling Rab35 GTPase.

This suggests that a different mechanism beyond the direct involvement of classical recycling pathways could be at play. Following the Reviewer’s suggestion, we have now omitted the colocalization studies between chased VE-cadherin and Rab4-GFP or LAMP2. Instead we have included a new set of experiments that shows that VE-cadherin internalization is increased in the absence of PACSIN2 or EHD4 (Fig. 6a, b), which supports the claim that the PACSIN2/EHD4 complex controls VE-cadherin trafficking. In addition, we decided to perform live imaging experiments using VE-cadherin-GFP/mCherry that visualize how EHD4 controls the local turnover of VE-cadherin at asymmetric AJ (New Fig. 6c and; New Supplementary Movie 3 and 4):

“Live imaging of HUVECs expressing EHD4-GFP and VE-cadherin-mCherry showed that the recruitment of EHD4 relates to rapid movements of the asymmetric AJs (Fig. 6c and Supplementary Movie 3). To investigate whether EHD4 controls local turnover of the AJs, VE-cadherin-GFP was expressed in shControl and shEHD4 HUVECs. Subsequent live cell imaging experiments clearly showed that the asymmetric AJs are gradually turning over during their movement through subtle internalization events (Fig. 6d and Supplementary Movie 4). Importantly, upon the depletion of EHD4, the asymmetric AJ are not turning over in a smooth fashion, but elongate instead until they break (Fig. 6d and Supplementary Movie 4). This junctional defect in shEHD4 HUVECs likely underlies the accumulation of VE-cadherin-positive vesicles in the pulse-chase experiments. “

Together, these insightful findings indicate that the PACSIN2/EHD4/MICAL-L1 complex mediates proper VE-cadherin trafficking at asymmetric junctions. The underlying mechanism of how PACSIN2 and EHD4 ensures local proper trafficking of VE-cadherin, and possibly recycling, is an interesting topic for future studies, in the revised manuscript we have toned down our statements regarding their probable role in switching between recycling and degradation pathways.

5) Along with the comments above, a model diagram describing how the pascin2 complex is regulating VE-cad trafficking would be helpful.

As suggested by the Reviewer, we have added a working model for the trafficking of VE-cadherin at asymmetric junctions (New Fig. 8).

Minor comments: We thank the reviewer for these helpful comments.

a) Secondary antibodies and catalog numbers are not mentioned in methods.

We have now specified this information in the updated reporting summary statement.

b) Company/ catalog no. of Mowiol4-88/DABCO is not mentioned.

Corrected

c) In the Fig. 1d legend, colors for ERG and IB4 are reversed, although correct in the figure itself.

Corrected

d) Fig. 4h, slight formatting error with black outline for F-actin.

Corrected

e) Fig. 7f, ERG is blue not green (in the legend).

Corrected

f) On page 7, switch Suppl. Figures 5 and 6 (either the numbers or the figures itself) to maintain continuity.

We swapped the figures as suggested.

Reviewer #2 (Remarks to the Author):

Collective cell migration is crucial for sprouting angiogenesis. During this process, coordinated rearrangement of ECs-ECs junctions mediated by VE-cadherin is important for the transmission of forces between leading cells and following cells. The authors have previously shown that PACSIN2, the F-BAR protein, is the key regulator of VE-cadherin at the tensile adherens junctions of the trailing ends of the migrating ECs in the monolayer (Dorland et al Nat. Commun 2016). Here in this manuscript, the authors generated PACSIN2 KO mice and analyzed the effect of loss of PACSIN2 during sprouting angiogenesis. The authors showed that PACSIN2 coordinates endothelial front-rear polarity in vivo and in vitro. Furthermore, the authors identified that EHD4 and MICAL-L1 forms the protein complex with PACSIN2 to modulate VE-cadherin trafficking in vitro. Finally, the authors confirmed that EHD4 KO mice shows a similar phenotype to PACSIN2 KO mice. The authors are advancing the concept that the molecular mechanisms underlying the transmission of tension at the rear of leading cells is important for the establishment of a front of migrating cells in vivo. Most of experiments were carefully designed and well done. The manuscript is written well and easy to follow. However, I feel the current experimental setting is not sufficient to draw the authors' conclusion. To improve this, the following points should be addressed

We kindly thank the Reviewer for the positive words and thorough analysis of the angiogenesis experiments. We have addressed the remaining points as follows:

Specific comments:

1. The analysis and interpretation of the phenotype of PACSIN2 KO mice at angiogenic front is unclear and insufficient. Are the shorter sprouting and clusters of ECs due to the defected polarized cell migration? Is the differentiation or gene transcription of ECs affected? For instance, is the expression of the tip cell marker OK?

To answer this question, we performed new imaging experiments on retinas isolated from newborn PACSIN2 knockout mice and stained the postnatal retina for the tip cell marker ESM1 together with the vascular markers ERG/IB4. These experiments show that the tip cells are still formed in the absence of PACSIN2 at the same ratio as in control retinas. This indicates that the EC differentiation programs are still fine, and strengthens the idea that the observed angiogenic defects are due to differences in endothelial dynamics during sprouting. We

incorporated these new data and insight in the revised manuscript (Fig. 1d and New Supplementary Fig. 2a). We also assessed whether the *Pacsin2*^{-/-} phenotype is due to defected polarized cell migration in vivo (see details at point 2).

2. Mechanical forces are important for asymmetric behavior of ECs. In vivo, endothelial cells are subjected to shear stress, which triggers directional endothelial migration (Franco et al., Plos Biol 2015). Additionally, ECs migrate not only toward the avascular region, even at the angiogenic front (Pitulescue et al Nat Cell Biol 2017). In figure 2, the author examined the effect of KD on Golgi orientation during the wound healing process. The Golgi orientation in ECs is very good indicator of polarized migration. Is the Golgi orientation affected in the PACSIN2 and EHD4 KO mice in regard to the polarized migration?

We agree with the Reviewer that assessment of Golgi orientation in vivo would show whether the vascular defects at the sprouting front of retinas in PACSIN2 or EHD4 knockout mice are due to differences in polarized EC migration, as our in vitro data implies. To address this point, we bred new mice, isolated retinas and stained P6 control and *Pacsin2*^{-/-} retinas for the Golgi marker GOLPH4 together with ERG (EC nuclei) and ICAM2 (vasculature) (Fig. 2m). These experiments showed that *Pacsin2*^{-/-} endothelial cells at the sprouting front fail to properly polarize towards the angiogenic front to support the formation of the new sprouts. This indicates that the lack of PACSIN2 indeed perturbs polarized collective migration in vivo (Fig. 2n, o), which strengthens our in vitro observations that already point towards an important role for PACSIN2 and EHD4 in endothelial collective migration. Due to the emerged Covid situation, which closed the Plomann lab for a long time, we are forced to reduce the size of mouse colonies and we were not able to collect more retinas to perform these experiments with *EHD4*^{-/-} retinas. Hence, we used the newly added *Pacsin2*^{-/-} data to support the claim that the PACSIN2-based complex is needed for polarized migration.

3. Is the barrier function of EC junction affected by PACSIN2 KO mice?

Based on the notion that we previously showed that depletion of PACSIN2 from in vitro cultured cells reduces endothelial barrier function, one may have expected the blood vessels in PACSIN2 KO mice to become leaky. However, the PACSIN2 KO mice develop until adulthood without major defects or bleedings, which indicates that there is no prominent vascular barrier defect. In accordance, our preliminary data further shows that no vascular leakage occurs of tail-vein injected Evans Blue in unchallenged mice (Fig. ii for Reviewers). There might be various explanations for this, including the compensatory mechanisms through the closely related PACSIN3 isoform or other F-BAR proteins (see also point 7). Because the current manuscript is not focusing on PACSIN2's role in vascular barrier function, and in accordance with the editorial recommendations, we have not further pursued experiments with the PACSIN2 KO mice to analyze its vascular barrier properties and function in greater detail and focused on developmental angiogenesis instead. We now mention in the corresponding results section: "*Pacsin2*^{-/-} mice are viable, fertile and adult mice appear healthy without major defects or bleedings, which indicates that there is no prominent vascular barrier defect."

Evans Blue injections

Figure ii: Images of brain and liver tissues from control and *Pacsin2*^{-/-} mice following tail-vein injections with Evans Blue.

4. Collective migration is a team effort for many cells. The question is whether the effect of PACSIN2 KD is cell-autonomous or not. Mosaic experiment *in vitro* would be interesting.

To decipher whether the effect of PACSIN2 knockdowns on collective migration is cell-intrinsic, we performed competition scratch assays. First, mosaic endothelial monolayers were generated in which half of the population of HUVECs expressed shControl with a RFP tag or HUVECs expressing shPACSIN2 and GFP. Next scratch assays were performed and the identity of the leader cells during collective migration was determined at t=0 and t=12 hours after scratch. The experiments demonstrated that the leading front is dominated by cells that express PACSIN2, whereas PACSIN2-depleted cells fail to follow during the collective cell migration process. This new data is added in the revised manuscript (New Fig. 2 g-i).

5. Where is PACSIN2 expressed *in vivo*? Are they specifically expressed at EC junction sites? Given the fact that the authors used global KO mice, this information is necessary. Additionally, if they use EC specific PACSIN2 KO mice, do they observe the similar phenotype with global KO mice?

We have previously shown that the PACSIN2 isoform is ubiquitously expressed in various tissues (Modregger et al 2000). For the *in vivo* experiments we used homologous recombination and the Cre-loxP recombination to create mice with a loxP-flanked (floxed) expression of the *Pacsin2* and *EHD4* genes. To generate animals lacking PACSIN2 or EHD4 expression, floxed mice were bred to transgenic Cre-deleter mice to produce knockout animals. Both mouse lines develop until adulthood and do not display critical defects in epithelial and mesenchymal tissues. Yet we do find differences specifically in the endothelium of newly developing blood vessels, suggesting that PACSIN2 and EHD4 play a key role in this tissue. Importantly, while we are aware of the limitations of the *in vivo* models, the observed vascular defects are strongly supported by the endothelial-specific PACSIN2 and EHD4 knockdown experiments *in vitro*. We were considering generating EC-specific PACSIN2 mouse models prior to the COVID situation, which has resulted in the shutting down of our mouse facility to new research projects until this day. Yet even if this was not the case, developing this model for experiments would take at the very least another 1.5 years. Whether the specific knockout of PACSIN2 from the endothelial cell lineage displays similar defects in angiogenesis *in vivo* is something we are keen to explore but have no information on at present. We have made a comment on this in the final future perspective sentence of the discussion in the revised manuscript.

To provide additional information regarding the expression of PACSIN in retinal tissue, we have stained wild-type P6 retinas for PACSIN2 and PACSIN3 (New Supplementary Fig. 1a), which confirmed that PACSIN2 is expressed in the retinal vasculature, whereas we detected no signal for PACSIN3. These experiments did not show prominent recruitment of PACSIN2 to junctions in the retinal vasculature. This may be due to the fact that the epitopes that this antibody recognizes differs from the one we use for immunofluorescence stainings of the

asymmetric junction between human endothelial cells, or potentially due to the notion that asymmetric AJs are very dynamic structures and PACSIN2-positive asymmetric AJs remodeling occurs rapidly, precluding their detection in fixed retinas. So currently, we do not know when PACSIN2 is recruited to endothelial junction during retinal angiogenesis. We have also stained P6 wild-type retinas for EHD4. In this case, we observed that EHD4 is highly expressed in the retinal vasculature and specifically enriched at endothelial junctional areas (Supplementary Fig. 8). The fact that EHD4 is not expressed at other retinal compartments and its deletion leads to a similar phenotype as the observed in *Pacsin2*^{-/-} retinas supports an endothelial role for the PACSIN2-EHD4 complex. The PACIN2, PACSIN3 and EHD4 stainings of the mouse retina have all been included as new data in the revised manuscript (New Suppl Fig. 1a and 8a).

6. In their previous manuscript, the authors revealed the role of PACSIN2 on VE-cadherin tensile junction, which is asymmetrically existing migrating ECs. While VE-cadherin is localized at all endothelial cell-to-cell contact sites, does PACSIN2 colocalize with VE-cadherin asymmetrically in vivo?

Please see our answer at point 5. In addition, we apparently forgot to mention that we previously showed that PACSIN2 is recruited to asymmetric junctions in vivo in human adult blood vessels (Figure iii for the Reviewers). Based on the Reviewer's question we now incorporated the in vivo existence of asymmetric junctions in the vasculature in the introduction: "The previous studies were performed with *in vitro* endothelial cell cultures, and although PACSIN2-positive asymmetric junctions have been observed in human blood vessels as well, to this date, the functional importance of junction-based signaling through BAR proteins in endothelial collective behavior and vascular development remains unknown".

Figure iii: En face confocal images of human mesenteric artery stained ex vivo for VE-cadherin and PACSIN2. This figure was previously published in Dorland et al. 2016 as Fig. 1f.

7. It seems that PACSIN3 expression is increased in the retina of PACSIN2 KO mice. Does this cause functional redundancy and rather weak phenotype of PACSIN2 KO mice? Is PACSIN3 expression increased in ECs?

Of the other PACSIN isoforms, PACSIN1 is not expressed in the endothelium, but PACSIN3 is. PACSIN3 is not recruited to endothelial junctions in the normal presence of its PACSIN2 counterpart (Dorland et al, 2016). Indeed, it is a realistic possibility that PACSIN3 compensates for the loss of PACSIN2 in the vasculature of PACSIN2 KO mice (see Supplementary Fig. 1e). To address that option, we have bred heterozygous *PACSIN2*^{+/-} and *PACSIN3*^{+/-} to generate double knockout mice. The pups of the breedings never resulted in PACSIN2 and PACSIN3 double knockouts, as they turn out to be embryonic lethal (Fig iv. for Reviewers).

Pascin2/3 breeding statistics		Total no. of littermates	Nos. per littermate genotype								
Parents genotype			P2+/- P3+/-	P2+/- P3+/-	P2+/- P3+/-	P2+/- P3+/-	P2+/- P3+/-	P2+/- P3+/-	P2+/- P3+/-	P2+/- P3+/-	
male	female										
P2 +/- P3 +/-	P2 +/- P3 +/-	148	18	31	0	14	62	14	8	14	0
		100%	12.2%	20.9%	-	9.5%	41.9%	9.5%	5.4%	9.5%	-
P2 -/- P3 +/-	P2 -/- P3 +/-	39	0	0	0	0	0	0	7	32	0
		100%	-	-	-	-	-	-	17.9%	82.1%	-
P2 +/- P3 -/-	P2 +/- P3 -/-	11	0	0	0	0	7	0	0	4	0
		100%	-	-	-	-	63.6%	-	-	36.3%	-
P2 -/- P3 +/-	P2 +/- P3 -/-	2	0	0	0	0	2	0	0	0	0
		100%	-	-	-	-	100%	-	-	-	-

Figure iv: Breeding statistics of PACSIN2 and PACSIN3 heterozygous knockout mice.

These experiments further demonstrate that the PACSIN proteins serve crucial developmental functions. However, because of the embryonic lethality of the double knockouts, this precluded us from studying retinal angiogenesis in the absence of both PACSIN isoforms simultaneously. Of note, knocking down expression of PACSIN2 from endothelial cells in vitro, does not lead to compensation of PACSIN3 (Fig v. for Reviewers)

Figure v: Western blot analysis in shControl vs. shPACSIN2 HUVECs.

8. In figure 1d, staining intensity against VE-cadherin seems to be increased not only in the cytoplasmic region but at cell-cell contact sites. This is not mentioned.

We apologize for that, we do not commonly see increased VE-cadherin intensity in the PACSIN2 knockout retinas. In the specific image the Reviewer referred to, there was a general increase of all markers in the immunostaining, which is a technical issue independent of analyzed genotype. We have now replaced this figure panel with a representative image (Fig. 1e).

9. Is the level of cytoplasmic VE-cadherin increased in EHD4 KO mice?

This is an interesting question and hence we have assessed the level of cytoplasmic VE-cadherin in control and *Ehd4*^{-/-} retinas. However, as we have previously explained, the Covid situation has a major impact on the amount of in vivo experiments that we could perform. We analyzed two different littermates with a total amount of 4 retinas per genotype. We observed a slight increase in the cytoplasmic VE-cadherin levels in the *Ehd4*^{-/-} vasculature compared to control retinas, similarly as the observed effect in the *Pascin2*^{-/-} retinas. The new VE-cadherin stainings in *Ehd4*^{-/-} retinas are added to the manuscript (New Fig. 7j) and although the levels are not statistically different, we have added this new quantified dataset to the revised manuscript (New Fig. 7k).

10. Is the increased level of cytoplasmic VE-cadherin observed in vitro as well?

Yes there is increased internalized VE-cadherin in vesicles upon the depletion of PACSIN2 or EHD4 in vitro (see new Fig 6a, b) and further explanation of these experiments at our answer to point 1 of Reviewer 1.

11. In Supplementary Fig1d, Ponceau S staining suggests the protein loading for western blotting analysis is not same.

Thank you for pointing this out. We performed new Western blotting on equally loaded samples. This analysis confirmed that PACSIN2 is depleted from the respective tissues, and that the *Pacsin2* gene deletion resulted in a slight increase in PACSIN1 and PACSIN3 protein levels in the overall *Pacsin2*^{-/-} retinal tissue (Supplementary Fig. 1e). Please note that PACSIN1 is not expressed in endothelial cells (Dorland et al., 2016). We have replaced the previous blots with these new data in Supplementary Fig. 1e and changed the text in the results section accordingly.

References

- Bentley, K., C.A. Franco, A. Philippides, R. Blanco, M. Dierkes, V. Gebala, F. Stanchi, M. Jones, I.M. Aspalter, G. Cagna, S. Weström, L. Claesson-Welsh, D. Vestweber, and H. Gerhardt. 2014. The role of differential VE-cadherin dynamics in cell rearrangement during angiogenesis. *Nat. Cell Biol.* 16:309–321. doi:10.1038/ncb2926.
- Cao, J., M. Ehling, S. März, J. Seebach, K. Tarbashevich, T. Sixta, M.E. Pitulescu, A.-C. Werner, B. Flach, E. Montanez, E. Raz, R.H. Adams, and H. Schnittler. 2017. Polarized actin and VE-cadherin dynamics regulate junctional remodelling and cell migration during sprouting angiogenesis. *Nat. Commun.* 8:2210. doi:10.1038/s41467-017-02373-8.
- Carvalho, J.R., I.C. Fortunato, C.G. Fonseca, A. Pezzarossa, P. Barbacena, M.A. Dominguez-Cejudo, F.F. Vasconcelos, N.C. Santos, F.A. Carvalho, and C.A. Franco. 2019. Non-canonical Wnt signaling regulates junctional mechanocoupling during angiogenic collective cell migration. *Elife.* 8. doi:10.7554/eLife.45853.
- Dorland, Y.L., T.S. Malinova, A.-M.D. van Stalborch, A.G. Grieve, D. van Geemen, N.S. Jansen, B.-J. de Kreuk, K. Nawaz, J. Kole, D. Geerts, R.J.P. Musters, J. de Rooij, P.L. Hordijk, and S. Huvneers. 2016. The F-BAR protein pacsin2 inhibits asymmetric VE-cadherin internalization from tensile adherens junctions. *Nat. Commun.* 7:12210. doi:10.1038/ncomms12210.
- Nanes, B.A., C.M. Grimsley-Myers, C.M. Cadwell, B.S. Robinson, A.M. Lowery, P.A. Vincent, M. Mosunjac, K. Früh, and A.P. Kowalczyk. 2017. p120-catenin regulates VE-cadherin endocytosis and degradation induced by the Kaposi sarcoma-associated ubiquitin ligase K5. *Mol. Biol. Cell.* 28:30–40. doi:10.1091/mbc.e16-06-0459.
- Modregger, J., B. Ritter, B. Witter, M. Paulsson, and M. Plomann. 2000. All three PACSIN isoforms bind to endocytic proteins and inhibit endocytosis. *J. Cell. Sci.* 24:4511-21.

REVIEWERS' COMMENTS

Reviewer #1 (Remarks to the Author):

The authors have done an outstanding job in responding to my comments on the initial submission. Overall, there is further clarity in the conclusions and discussion, additional supportive data, and new diagrams that clarify the authors proposed mechanisms. I applaud the authors for a nice study and do not have further comments or suggestions.

Reviewer #2 (Remarks to the Author):

The authors addressed most of my comments and another reviewer's comment. Due to the current covid19 situation, they could not address in vivo endothelia polarization in the EHD4 KO mice. However, they examined in vivo endothelial polarization with the Pacin2 KO mice. As the Pacin2 KO mice exhibited defected EC polarity, I feel the authors' in vitro observation was supported by in vivo. The manuscript was improved and now ready for publication by Nature communications.